# BPC 157 Therapy and the Permanent Occlusion of the Superior Sagittal Sinus in Rat: Vascular Recruitment

**DOI:** 10.3390/biomedicines9070744

**Published:** 2021-06-28

**Authors:** Slaven Gojkovic, Ivan Krezic, Hrvoje Vranes, Helena Zizek, Domagoj Drmic, Katarina Horvat Pavlov, Andrea Petrovic, Lovorka Batelja Vuletic, Marija Milavic, Suncana Sikiric, Irma Stilinovic, Mariam Samara, Mario Knezevic, Ivan Barisic, Ivica Sjekavica, Eva Lovric, Anita Skrtic, Sven Seiwerth, Predrag Sikiric

**Affiliations:** 1Departments of Pharmacology, School of Medicine, University of Zagreb, 10 000 Zagreb, Croatia; slaven.gojkovic.007@gmail.com (S.G.); ivankrezic94@gmail.com (I.K.); hrvoje.vranes@gmail.com (H.V.); zizekhelena@gmail.com (H.Z.); iddrmic@mef.hr (D.D.); stilinovic.i@gmail.com (I.S.); mariam.samara@hotmail.com (M.S.); mariknezevic@gmail.com (M.K.); inbarisic@gmail.com (I.B.); 2Departments of Pathology, School of Medicine, University of Zagreb, 10 000 Zagreb, Croatia; katarina.horvat@gmail.com (K.H.P.); petrovic.andrea@gmail.com (A.P.); lbatelja@mef.hr (L.B.V.); marija.milavic@mef.hr (M.M.); suncanasikiric@gmail.com (S.S.); eva.lovric@kb-merkur.hr (E.L.); seiwerth@mef.hr (S.S.); 3Department of Diagnostic and Interventional Radiology, University Hospital Centre, 10 000 Zagreb, Croatia; Ivica.sjekavica@zg.t-com.hr

**Keywords:** BPC 157, superior sagittal sinus, occlusion, therapy, vascular recruitment, rats

## Abstract

We show the complex syndrome of the occluded superior sagittal sinus, brain swelling and lesions and multiple peripheral organs lesions in rat. Recovery goes centrally and peripherally, with the stable gastric pentadecapeptide BPC 157, which alleviated peripheral vascular occlusion disturbances, rapidly activating alternative bypassing pathways. Assessments were gross recording, venography, ECG, pressure, microscopy, biochemistry. The increased pressure in the superior sagittal sinus, portal and caval hypertension, aortal hypotension, arterial and venous thrombosis, severe brain swelling and lesions (cortex (cerebral, cerebellar), hypothalamus/thalamus, hippocampus), particular veins (azygos, superior mesenteric, inferior caval) dysfunction, heart dysfunction, lung congestion as acute respiratory distress syndrome, kidney disturbances, liver failure, and hemorrhagic lesions in gastrointestinal tract were all assessed. Rats received BPC 157 medication (10 µg/kg, 10 ng/kg) intraperitoneally, intragastrically, or topically to the swollen brain at 1 min ligation-time, or at 15 min, 24 h and 48 h ligation-time. BPC 157 therapy rapidly attenuates the brain swelling, rapidly eliminates the increased pressure in the ligated superior sagittal sinus and the severe portal and caval hypertension and aortal hypotension, and rapidly recruits collateral vessels, centrally ((para)sagittal venous collateral circulation) and peripherally (left superior caval vein azygos vein-inferior caval vein). In conclusion, as shown by all assessments, BPC 157 acts against the permanent occlusion of the superior sagittal sinus and syndrome (i.e., brain, heart, lung, liver, kidney, gastrointestinal lesions, thrombosis), given at 1 min, 15 min, 24 h or 48 h ligation-time. BPC 157 therapy rapidly overwhelms the permanent occlusion of the superior sagittal sinus in rat.

## 1. Introduction

We attempt to overwhelm in rats the permanently occluded superior sagittal sinus and central venous occlusion [1,2,3,4] and resolve the consequences thereof (i.e., multiorgan failed syndrome). The resolving focus was on the stable gastric pentadecapeptide BPC 157 [5,6,7,8,9,10,11,12,13,14,15,16,17,18,19], its therapy effect shown to overwhelm peripheral venous occlusion syndromes [20,21,22,23,24,25,26,27]. The activated “bypassing key” was the rapid vessels recruitment; collateral pathways (i.e., left ovarian vein [20], inferior mesenteric vein [25], azygos vein [26]), reliant on the injurious occlusion, reestablished blood flow that compensated the vessel occlusion [20,21,22,23,24,25,26,27]. Illustratively, BPC 157 therapy attenuated/eliminated the whole Budd–Chiari syndrome [26] (i.e., left superior caval vein-azygos vein-inferior caval vein shunt to skip the suprahepatic occlusion of the inferior caval vein), including the counteraction of the prominent deadly syndrome. Heart dysfunction, lung lesions (i.e., time-dependent and time-independent features that can be acute respiratory distress syndrome exudative phase features), liver failure, and gastrointestinal lesions, widespread arterial and venous thrombosis, severe portal and caval hypertension and aortal hypotension were all counteracted [26]. Providing the closely interrelated increased pressure in the three body cavities interacting with each other rapidly transmitted through the venous system [28,29] (and thereby, venous system as therapy target [20,21,22,23,24,25,26,27]), the occluded superior sagittal sinus in addition to the brain swelling and lesions and increased pressure in the superior sagittal sinus may induce a comparable deadly syndrome. Previously, with BPC 157 therapy application, an alike therapy maneuver and beneficial effect appeared both in the Pringle maneuver ischemia and reperfusion [25]. Additionally, in the stroke-studies given after reperfusion initiation, after carotid arteries clamping, BPC 157 abrogated hippocampal ischemia/reperfusion injuries in rats [30]. Additionally, BPC 157 may participate in the brain–gut and gut–brain axis function [9], exerting particular effects when given peripherally [9] (i.e., release of the serotonin in the specific brain areas (i.e., nigrostriatum) [31], opposes the schizophrenia-like positive symptoms models [32], counteracts various encephalopathies [30,33,34,35,36,37,38,39,40,41]).

Accordingly, with BPC 157 therapy in the rats with the permanently occluded superior sagittal sinus, rapid upgrading of the bypassing pathways, both centrally and peripherally, would resolve vessel occlusion disturbances, both centrally and peripherally [20,21,22,23,24,25,26,27]. The study includes both the very early and prolonged periods. To cover the full injury course, both the acute and prolonged periods, the supporting evidence goes with the various time points (i.e., at 1 min, 15 min, 24 h or 48 h ligation-time) for the application of the BPC 157 therapy. Local application at the swollen brain means a direct effect; intraperitoneal or intragastric administrations mean a systemic effect, µg- and ng-regimens mean a common beneficial effect. Conceptually, intragastric application benefits BPC 157 importance as an original cytoprotective anti-ulcer peptide (i.e., epithelium, endothelium maintenance and protection) [5,6,7,8,9,10,11,12,13,14,15,16,17,18,19]. It is resistant and stable in the human gastric juice (more than 24 h [42]), acting as a membrane stabilizer, counteracting leaky gut syndrome, as a particular target [19] distinctive from the standard peptide growth factors [16] (i.e., standard peptide growth factors are rapidly destroyed in human gastric juice [42]), with particular molecular pathways involved [15,19,20,28,43,44,45,46,47,48,49,50]. Likely, BPC 157 is controlling VEGF- and NO-pathways [43,47]. BPC 157 also immediately triggered the internalization of VEGFR2 and subsequently activated the phosphorylation of VEGFR2, Akt, and eNOS signal pathway without the need of other known ligands or shear stress [47]. Finally, in addition to this particular effect [47], there is a direct effect on vasomotor tone (i.e., BPC 157 counteracts both L-NAME-induced hypertension and L-arginine-induced hypotension, and there is a specific activation of Src-Caveolin-1-endothelial nitric oxide synthase (eNOS) pathway [43]. There, the activated “bypassing key” appeared as an outbreak of the original cytoprotection agent’s activity of “the endothelium protection → epithelium protection” [1,7,8]. The additional equation “endothelium maintenance → epithelium maintenance = blood vessel recruitment and activation (“running”) towards the site of injury, also described as “bypassing” occlusion via alternative ways” [7,8], was shown in the BPC 157 therapy effect in the previous peripheral vascular occlusion studies [20,21,22,23,24,25,26,27].

Thus, this point should be useful in the central venous occlusion, and we performed further BPC 157 studies in the rats with the occluded superior sagittal sinus.

## 2. Materials and Methods

### 2.1. Animals

This study was conducted with 12-week-old, 200 g body weight, male albino *Wistar* rats, randomly assigned at 6 rats/group/interval. Rats were bred in-house at the Pharmacology animal facility, School of Medicine, Zagreb, Croatia. The animal facility was registered by the Directorate of Veterinary (Reg. No: HR-POK-007). Laboratory rats were acclimated for five days and randomly assigned to their respective treatment groups. Laboratory animals were housed in polycarbonate (PC) cages under conventional laboratory conditions at 20–24 °C, relative humidity of 40–70% and noise level 60 dB. Each cage was identified with dates, number of study, group, dose, number and sex of each animal. Fluorescent lighting provided illumination 12 h per day. Standard good laboratory practice (GLP) diet and fresh water was provided ad libitum. Animal care was in compliance with standard operating procedures (SOPs) of the Pharmacology animal facility, and the European Convention for the Protection of Vertebrate Animals used for Experimental and other Scientific Purposes (ETS 123).

This study was approved by the Ethic Committee of School of Medicine, University of Zagreb (Project code: 380-59-10106-17-1001290; Data of approval: 2 November 2020). Ethical principles of the study complied with the European Directive 010/63/E, the Law on Amendments to the Animal Protection Act (Official Gazette 37/13), the Animal Protection Act (Official Gazette 135/06), the Ordinance on the protection of animals used for scientific purposes (Official Gazette 55/13), Federation of European Laboratory Animal Science Associations (FELASA) recommendations and the recommendations of the Ethics Committee of the School of Medicine, University of Zagreb. The experiments were assessed by observers blinded as to the treatment.

### 2.2. Drugs

Medication was administered as described previously [20,21,22,23,24,25,26,27], without use of a carrier or peptidase inhibitor, for stable gastric pentadecapeptide BPC 157, a partial sequence of the human gastric juice protein BPC, which was freely soluble in water at pH 7.0 and in saline. BPC 157 (GEPPPGKPADDAGLV, molecular weight 1419; Diagen, Ljubljana, Slovenia) was prepared as a peptide with 99% high-performance liquid chromatography (HPLC) purity, with 1-des-Gly peptide being the main impurity. The dose and application regimens were as described previously [20,21,22,23,24,25,26,27].

### 2.3. Experimental Protocol

As a methodological improvement to the absolute obstruction of the superior sagittal sinus, the sinus was ligated to the bone while the skull bone overlying the sinus eliminates the risk of sinus tear. We made 2 burr holes in the anesthetized rats (intraperitoneal (ip) injected 40 mg/kg thiopental (Rotexmedica, Germany) and 10 mg/kg diazepam (Apaurin; Krka, Slovenia)), each approximately 2 mm laterally from the middle of the sagittal suture. Then, we used a 5-0 Vicryl suture (or a 4-0 non-absorbable silk suture) to make a ligation in the middle of the superior sagittal sinus. Furthermore, methodologically, the sinus ligated to the bone provides the absolute obstruction of the sinus while skull bone overlying the sinus eliminates the risk of sinus tear. Thereby, permanent occlusion by ligation of the superior sagittal sinus leads to permanent alteration of blood flow, and permanent increased pressure in the superior sagittal sinus. Note, the described craniotomy procedure (including complete calvariectomy (see below)) alone was without any additional harmful effect (data not specifically shown).

For the rats euthanized at 15 min, 24 h and 48 h ligation-time, medication was at 1 min ligation-time, 10 µg/kg BPC 157, 10 ng/kg BPC 157, or 5 mL/kg saline. It was given (i) intraperitoneally or (ii) intragastrically or (iii) topically at the swollen brain through an additional midline burr hole placed approximately 2 mm rostral to the horizontal line connecting the two parasagittal burr holes, making a triangle with them and overlying the sagittal sinus in the dural.

For venography, medication (10 µg/kg BPC 157, 10 ng/kg BPC 157 or 5 mL/kg saline) was applied intraperitoneally, at 15 min ligation-time, just before venography. Special emphasis was made to perceive the effect on the additional hypertension inside the cranium, or in the neck or in the abdomen, produced to the pre-existing venous hypertension by venography volume application (i.e., additional volume inside the sinus lumen, or in the right external jugular vein, or in the inferior caval vein). Thereby, at 15 min ligation-time, just before the saline volume application (1 mL/rat) into the superior sagittal sinus, right external jugular vein or inferior caval vein, medication (10 µg/kg BPC 157, 10 ng/kg BPC 157 or 5 mL/kg saline) was applied intraperitoneally.

For the pressure recordings in the superior sagittal sinus, portal vein, vena cava, and abdominal aorta (throughout 5 min), medication (10 µg/kg BPC 157, 10 ng/kg BPC 157 or 5 mL/kg saline) (i) intraperitoneally or (ii) intragastrically or (iii) topically at the swollen brain through an additional midline burr hole. The time points were at 1 min ligation-time (early effect) or at 15 min, 24 h or 48 h ligation-time (delayed application). In the case of the additional hypertension made inside the cranium, neck and abdomen at 15 min ligation-time, medication was given intraperitoneally, just before saline volume (1 mL/rat) application into the superior sagittal sinus, right external jugular vein or inferior caval vein medication.

Recording of the brain swelling was performed in rats at 15 min after the complete calvariectomy was performed. Briefly, 6 burr holes were drilled in three horizontal lines, all of them medially to the superior temporal lines and temporalis muscle attachments. The rostral two burr holes were placed just basal from the posterior interocular line, the basal two burr holes were placed just rostral to the lambdoid suture (and transverse sinuses) on both sides, respectively, and the middle two burr holes were placed in the line between the basal and rostral burr holes. A laparotomy was made for the corresponding presentation of the peripheral veins (azygos, superior mesenteric, inferior caval) recording. Medication was 10 µg/kg BPC 157, 10 ng/kg BPC 157, or 5 mL/kg saline, given intraperitoneally, intragastrically, or as bath to the swollen brain above the sinus, at 15 min ligation-time, and recording (with a camera attached to a VMS-004 Discovery Deluxe USB microscope (Veho, Dayton, OH, USA)) performed through next 10 min time. Alternatively, the described medication was at 1 min ligation time, and recording (with a camera attached to a VMS-004 Discovery Deluxe USB microscope (Veho, Dayton, OH, USA)) was performed before sacrifice at 15 min, 24 h or 48 h ligation-time.

### 2.4. Venography

Venography was performed in rats with a ligation of the superior sagittal sinus at 15 min post-ligation, using a C-VISION PLUS fluoroscopy unit (Shimadzu, Chiyoda, Tokyo, Japan) [20,26,27]. One ml throughout 45 s warmed Omnipaque 350 (iohexol) non-ionic contrast medium (GE Healthcare, Arlington Heights, IL, USA) was injected into the superior sagittal sinus anterior to ligation, into the right external jugular vein and into the inferior vena cava at the level of bifurcation of rats with a ligated superior sagittal sinus. The contrast medium was visualized under real-time to ensure adequate filling. A subtraction mode was used to record the images at 14 frames per second. At 15 min post-ligation, venograms were taken, captured, and digitized into files on a personal computer and were analyzed using ISSA image software (Vamstec, Zagreb, Croatia). Venography assessment includes rats having a full presentation of collaterals (i.e., presentation of pterigopalatinal veins and nasal veins and presentation of the ophthalmic vein, angularis vein, facial anterior and posterior vein, and facial vein, or through cerebri superior veins, sinus cavernosus, sinus petrosus superior and inferior, sinus transversus, through jugular external vein, subclavia vein through superior caval veins after venography into the superior sagittal sinus; partial visualization of azygos vein, no sign of pulmonary congestion, normal heart contrast filling (venography into right external jugular vein); no congestion within hepatic veins with parenchymal liver phase, mild congestion in pulmonar artery with parenchymal lung phase, normal heart filling with contrast media (venography through inferior caval vein)).

### 2.5. Superior Sagittal Sinus, Portal and Caval Vein and Abdominal Aorta Pressure Recording

Mean blood pressure recordings were made in deeply anesthetized after with a cannula (BD Neoflon™ Cannula) connected to a pressure transducer (78534C MONITOR/TERMINAL; Hewlett Packard, Houston, TX, USA) inserted into the superior sagittal sinus, portal vein, inferior vena cava and abdominal aorta at the level of the bifurcation at 15 min, 24 h or 48 h post-ligation. Each recording lasted five minutes, being assessed in one-minute intervals. For superior sagittal sinus pressure recording, we made a single burr hole in the rostral part of the sagittal suture, above the superior sagittal sinus, and cannulated superior sagittal sinus anterior part by Braun intravenous cannulas, and then, we laparatomized rats for portal vein, inferior vena cava and abdominal aorta pressure recording.

Notably, normal rats exhibited a superior sagittal sinus pressure in a range from −24 to −27 mmHg, and a portal pressure of 3–5 mmHg similar to that of the inferior vena cava, though with at least 1 mmHg higher values in the portal vein. By contrast, abdominal aorta blood pressure values were 100–120 mm Hg at the level of the bifurcation [20,26,27].

### 2.6. ECG Recording

ECGs were recorded continuously in deeply anesthetized rats for all three main leads, by positioning stainless steel electrodes on all four limbs using an ECG monitor with a 2090 programmer (Medtronic, Minneapolis, MN, USA) connected to a Waverunner LT342 digital oscilloscope (LeCroy, Chestnut Ridge, NY, USA) at 15 min, 24 h or 48 h ligation-time. This arrangement enabled precise recordings, measurements and analysis of ECG parameters [20,26,27].

### 2.7. Histology

Tissue specimens from the brain, liver, spleen, stomach, duodenum, lungs and heart were obtained from rats with superior sagittal sinus ligation at 15 min (and 1, 5 and 10 min intervals thereafter), 24 h and 48 h post-ligation. These were fixed in buffered forma-lin (pH 7.4), for 24 h, dehydrated, and embedded in paraffin wax. The samples were stained with hematoxylin-eosin according to automated Sakura Tissue-Tek DRS 2000 Slide Stainer protocol (https://www.sakura.eu/Solutions/Staining-Coverslipping/H-E-Kit, accessed on 6 June 2021) following procedural steps: rehydration to distilled water; staining with haematoxylin; washing in running tap water; differentiation with 70% alcohol; staining with eosin; dehydration; clearing; mounting. Tissue injury was evaluated microscopically by a blinded examiner using Olympus BX51 microscope and Olympus 71 digital camera for saving images as uncompressed 24-bit RGB TIFF files. Specifically, the brains were dissected using coronal section with mandatory 2 sections according to NTP-7, Level 3 and 6, due to neuroanatomic subsites present in certain brain sections [51]. At NTP-7 Level 3 we observed area of fronto-parietal cortex, hippocampus, thalamus and hypothalamus. At NTP-7 Level 6 we analyzed cerebellar cortex morphology. Brain coronal blocks were embedded in paraffin, sectioned at 4 μm, stained with H&E and evaluated by light microscopy using neuropathological scoring.

Brain histology. Brain injury in different regions was evaluated using a semiquantitative neuropathological scoring system as described [52] (Table 1), providing a common score 0–8, grade 0 indicates no histopathologic damage. Semiquanitative scoring of the microglia recruitment was analyzed by a semiquantitative score combining both the number of cells and their morphology. As described [53] for the grey matter, the following scoring definitions were applied: score 1 = less than 5 cells; score 2 = between 5 and 15 cells of which less than 25% with an amoeboid morphology; score 3 = more than 15 cells of which more than 25% have an amoeboid morphology at magnification 600×.

Lung histology. The following scoring system to grade the degree of lung injury was used in lung tissue analysis. Features were focal thickening of the alveolar membranes, congestion, pulmonary edema, intra-alveolar hemorrhage, interstitial neutrophil infiltration, and intra-alveolar neutrophil infiltration. Each feature was assigned a score from 0 to 3 based on its absence (0) or presence to a mild (1), moderate (2), or severe (3) degree, and a final histology score was determined [54].

Renal, liver, heart histology. The renal injury was based on degeneration of Bowman space and glomeruli, degeneration of proximal and distal tubule, vascular congestion and interstitial edema. The criteria for liver injury were vacuolization of hepatocytes and pyknotic hepatocyte nuclei, activation of Kupffer cells and enlargement of sinusoids. Each specimen was scored using a scale ranging from 0–3 (0: none, 1: mild, 2: moderate, and 3: severe) for each criterion and a final histology score was determined [55]. The hearth lesion estimation was based with dilatation and congestion of blood vessels within myocardium and coronary arteries as present (scored 1) or no present (scored 0).

#### Immunohistochemistry

Sequentially sectioned 4 μm-thick slides were used for performing immunohistochemical staining of microglial cells in rats euthanized at 48 h. Immunohistochemical staining by an automated immunostainer (Dako Autostainer Plus, DakoCytomation, Glostrup, Denmark) was performed using LSAB HRP and HRP+ kits according to the manufacturer’s instructions. For establishing microglial cells, we used CD68, KP1 (clone KP1, monoclonal, antibody-online.com, dilution 1:200); and CD68 (PG-M1) (clone KP1, monoclonal, antibody-online.com, dilution 1:200), and CD163 (clone ED2, Thermo Fisher Scientific, CA, USA, dilution 1:200), for M1 and M2 macrophages, respectively.

### 2.8. Thrombus Assessment

On being euthanized, the superior sagittal sinus, and peripherally, in portal vein, inferior caval vein, superior mesenteric vein, lienal vein and abdominal aorta were removed from the rats, and clots were weighed [20,26,27].

### 2.9. Brain Volume and Vessels Presentation Proportional with the Change of the Brain or Vessels Surface Area

The presentation of the brain, and peripheral veins (azygos, superior mesenteric and inferior caval) was recorded in deeply anaesthetized rats, with a camera attached to a VMS-004 Discovery Deluxe USB microscope (Veho, USA), before procedure in normal, and then, in rats with ligated superior sagittal sinus 15 min after procedure, before and after therapy as well as at the 15 min, 24 h and 48 h ligation-time before sacrifice. The border of the brain or veins were photographed and marked using ImageJ computer software and then, the surface area (in pixels) of the brain or veins was measured using a measuring function. This was performed with brain photographs before the application and at intervals after the application for both control and treated animals. The brain or veins area before the procedure and application was marked as 100% and the ratio of each subsequent brain area to the first area was calculated ($\frac{A_{2}}{A_{1}}$). Starting from square-cube law Equations (1) and (2), an equation for change of brain volume proportional with the change of the brain surface area (6) was derived. In Expressions (1)–(5) *l* is defined as any arbitrary one-dimensional length of brain (for example rostro-caudal length of the brain); used only for defining one dimensional proportion (*l_2_/l_1_*) between two observed brains and as an inter-factor (and because of that not measured (6)) for deriving the final expression [6]. The procedure was as follows: $A_{2}=A_{1}\times {\left(\frac{l_{2}}{l_{1}}\right)}^{2}$ (1) (square-cube law), $V_{2}=V_{1}\times {\left(\frac{l_{2}}{l_{1}}\right)}^{3}$ (2) (square-cube law), $\frac{A_{2}}{A_{1}}={\left(\frac{l_{2}}{l_{1}}\right)}^{2}$ (3) (from (1), after dividing both sides by A_1_), $\frac{l_{2}}{l_{1}}=\sqrt{\frac{A_{2}}{A_{1}}}$ (4) (from (3), after taking square root of both sides), $\frac{V_{2}}{V_{1}}={\left(\frac{l_{2}}{l_{1}}\right)}^{3}$ [5] (from (2), after dividing both sides by V_1_), $\frac{V_{2}}{V_{1}}={\left(\sqrt{\frac{A_{2}}{A_{1}}}\;\right)}^{3}$ (6) (after incorporating expression (4) into Equation (5)).

### 2.10. Stomach, Duodenum, Liver, Spleen, Ascites Presentation

The presentation of the gross lesions in gastrointestinal tract and vessels was recorded in deeply anaesthetized rats, with a camera attached to a VMS-004 Discovery Deluxe USB microscope (Veho, USA). At 15 min, 24 h and 48 h post-ligation, we assessed hemorrhagic congestive areas in the stomach and duodenum, (sum of the longest diameters, mm). The assessments occurred at selected time points before and after therapy, in rats with a suprahepatic occlusion of the inferior vena cava, at 5, 10 and 15 min post-ligation. Liver and spleen weights were expressed as a percent of the total body weight (for normal rats, liver 3.2–4.0% and spleen 0.20–0.26%). Ascites (mL) was also assessed.

### 2.11. Bilirubin and Enzyme Activity

To determine the serum levels of aspartate transaminase (AST), alanine transaminase (ALT, IU/L), and total bilirubin (μmol/L), blood samples were collected immediately after euthanasia and were centrifuged for 15 min at 3000 rpm. All tests were performed using an Olympus AU2700 analyzer with original test reagents (Olympus Diagnostics, Ireland) [24]. However, since there was no increase in bilirubin, the data were not shown.

### 2.12. Statistical Analysis

Statistical analysis was performed by parametric one-way analysis of variance (ANOVA), with post-hoc Newman–Keuls test and non-parametric Kruskal–Wallis test and subsequently the Mann–Whitney U test to compare groups. Values were presented as the mean ± standard deviation (SD) and as the minimum/median/maximum. To compare the frequency difference between groups, the chi-square test or Fischer’s exact test was used. *p* < 0.05 was considered statistically significant.

## 3. Results

We described BPC 157 therapy in rats that overwhelms the occluded superior sagittal sinus and consequent major syndrome, centrally and peripherally. Likely, this may be by rapidly activating alternative bypassing pathways. Consistent gross recording, venography, ECG, pressure, microscopy and biochemistry assessment clearly described the aggressive syndrome of the occluded superior sagittal sinus. There is the increased pressure in the superior sagittal sinus presenting with the systemic arterial and venous thrombosis, severe brain and peripheral disturbances, cardiac dysfunction, lung congestion, kidney congestion, liver failure, portal and caval hypertension, aortal hypotension and hemorrhagic lesions in gastrointestinal tract, especially in the stomach and duodenum. The therapy implies the full injury course, for both acute and prolonged periods, early and delayed application (i.e., at 1 min, 15 min, 24 h or 48 h ligation-time). BPC 157 therapy was effective when given topically at the swollen brain, intraperitoneally and intragastrically, µg- and ng-regimens.

### 3.1. Brain Swelling and Vessels Presentation and Counteraction

The occluded superior sagittal sinus rapidly develop brain swelling in rat (brain volume proportional with the change of the brain surface area reveals an immediate increase to the 140% over the initial presentation) and peripheral vessels failure (Figure 1, Figure 2, Figure 3 and Figure 4). As a consistent and prominent effect, BPC 157 therapy rapidly attenuates the brain swelling close to the normal, pre-procedure values. The consistent therapy evidence shows the effect of the local application at the swollen brain, but also an alike effect with the intraperitoneal and intragastric, µg- and ng-regimens. This effect is parallel with the effect on the peripheral blood vessels presentation. As a vein running between the left superior caval vein and inferior caval vein, the azygos vein presentation was reversed from the thin to the exaggerated presentation along with BPC 157 application, as seen at 15 min (and close period thereafter), 24 h, 48 h ligation time. Likewise, the inferior caval vein and superior mesenteric vein, congested presentation was rapidly reversed to normal vessel presentation upon BPC 157 administration, given as an abdominal bath (intraperitoneally), as a bath to the swollen brain, or intragastrically.

### 3.2. Superior Sagittal Sinus, Portal and Caval Vein and Abdominal Aorta Pressure Recording

Ligation of the superior sagittal sinus immediately overwhelms normal (negative) pressure, and induces the increased (positive) pressure, along with the severe portal and caval hypertension (portal hypertension exceeding caval hypertension), and aortal hypotension, persisting throughout the entire experimental period (Figure 5). With BPC 157 therapy, given as local bath at the swollen brain, but also intraperitoneal and intragastric application, µg- and ng-regimens, at 1 min, 15 min, 24 h or 48 h ligation-time, the increased (positive) pressure in the superior sagittal sinus is immediately reversed to the normal (negative) pressure. Elimination of the severe portal and caval hypertension rapidly appears. Aortal hypotension is rapidly resolved. Thus, it likely means adequate resolving of the anatomical imbalance in venous drainage, ongoing or pre-existing.

### 3.3. Thrombosis

Thrombosis rapidly appears (i.e., 15 min), progressing both centrally, in the superior sagittal sinus, and peripherally, in portal vein, inferior caval vein, superior mesenteric vein, lienal vein and abdominal aorta (Figure 6). BPC 157, given at 1 min ligation time, or later, at 15 min, 24 h or 48 h markedly counteracted and reversed thrombosis presentation.

### 3.4. Venography in the Superior Sagittal Sinus, Right External Vein, Inferior Caval Vein

Without medication, rats having the occluded superior sagittal sinus regularly show poor presentation in the venography (1 mL through 30 s in the superior sagittal sinus, right external jugular vein, inferior caval vein) (Figure 7). Commonly, they respond with the apparent lack of the activated collaterals. This occurs upon the additional pressure increase of the already severe superior sagittal sinus, portal and caval hypertension (venous hypertension additionally originated within cranium (superior sagittal sinus venography)). Likewise, this is also with the additional pressure increase in the superior sagittal sinus (where the severe portal and caval hypertension were not further raised by venous hypertension additionally originated within neck (venography into the right jugular vein), or abdomen (venography into the inferior caval vein)). Commonly, there are pulmonary congestion, heart dilatation and insufficient heart contrast filling (venography in the superior sagittal sinus, right external jugular vein), poor inferior caval vein compliance, huge congestion within prominent hepatic veins without parenchymal liver phase (venography through inferior caval vein).

BPC 157 medication fully counteracts these disturbances (*p* ˂ 0.05, at least, vs. control). Consistent with the evidenced counteraction of the increased pressure in the superior sagittal sinus, portal and caval hypertension, and the counteraction of the additional venous hypertension, either originated centrally (inside the sinus lumen) or peripherally (into the neck or abdomen), venography in all BPC 157 rats reveals activated collaterals presentation. Venography in the superior sagittal sinus and in the right jugular external vein reveals bypassing of the termination of the superior sagittal sinus with collaterals presentation centrally and peripherally. There is the ophthalmic vein, angularis vein, facial anterior and posterior vein, and facial vein, or through cerebri superior veins, sinus cavernosus, sinus petrosus superior and inferior, sinus transversus, through the jugular external vein, subclavia vein through the superior caval vein, retrograde filling of both jugular external veins and both superior caval veins with vertebral veins. Peripherally, there are the azygos vein, retrograde filling of dilated inferior caval vein with paravertebral venous plexus, slight visualization of hepatic veins inflow, and no sign of pulmonary congestion and heart dilatation and normal filling of contrast media in the heart. BPC 157 venography through the inferior caval vein goes without congestion within hepatic veins with parenchymal liver phase.

### 3.5. Pressure Recording Upon Additional Venous Hypertension Originated Inside the Sinus Lumen, Neck and Abdomen

We particularly investigated the additional hypertension made inside the cranium, neck and abdomen at 15 min ligation-time (Figure 5). BPC 157 or saline medication was given intraperitoneally, just before saline volume (1 mL/rat) application into the superior sagittal sinus, right external jugular vein or inferior caval vein.

An additional volume application into the sinus immediately aggravates a raise of the venous hypertension that additionally originated within the cranium, inside the sinus lumen as well as the portal hypertension and caval hypertension further affected. As before (Figure 5), BPC 157-rats maintain the rapid elimination of the increased pressure in the ligated superior sagittal sinus as well as the elimination of the portal/caval hypertension and aortal hypotension, and noteworthy, the BPC 157-rats show the rapid presentation of the shunts from the superior sagittal sinus.

Likewise, with application of the additional volume in the right external jugular vein or in the inferior caval vein, the elimination of the increased pressure in superior sagittal sinus, portal and caval hypertension, and aortal hypotension may act against the additional peripheral challenge. There is the elimination of the venous hypertension additionally originated in the neck or in the abdomen. Otherwise, there is the milder increases of the pressure in the ligated superior sagittal sinus upon additional volume application in the right external jugular vein or in the inferior caval vein, where in the control rats, the portal/caval hypertension remained unchanged, but contributed to the further raise of the increased pressure in the superior sagittal sinus (Figure 5).

### 3.6. Stomach, Duodenum, Liver, Spleen, Ascites Presentation

Occluded superior sagittal sinus regularly induces in rats the stomach and duodenum hemorrhagic lesions, and considerable ascites presentation, which were largely counteracted with BPC 157 therapy application (Figure 4; Figure 8). Liver and spleen weight markedly increased at the 24 h and 48 h period, but not in BPC 157 treated rats (Figure 2).

### 3.7. ECG Recording

Regularly, ECGs recording show severe tachycardia and prolonged QT-interval at 15 min, 24 h or 48 h ligation-time, which were markedly counteracted by BPC 157 regimens (Figure 2).

### 3.8. Enzymes

Serum ALT and AST values increased in the controls; they were lower in rats treated with BPC 157 at all-time intervals (Figure 8).

### 3.9. Histology

To support BPC 157 therapy which may overwhelm in rats the occluded superior sagittal sinus, we presented consistent findings (i.e., brain swelling and vessels presentation and counteraction, superior sagittal sinus, portal and caval vein and abdominal aorta pressure recording, thrombosis, venography, stomach, duodenum, liver, spleen, ascites presentation, ECG recording and enzymes presentation). The corresponding microscopic evidence in the brain and periphery (Figure 9, Figure 10, Figure 11, Figure 12, Figure 13, Figure 14, Figure 15 and Figure 16) is likewise. Together, they fully support a particular syndrome in the ligated superior sagittal sinus rats. This may be ongoing Virchow’s triad centrally and peripherally, and counteracting potential of the given BPC 157 therapy.

Specifically, after ligation, in the control rats, the brain damages are widespread, with increased edema and congestion, rapidly and particularly progressing as shown in the cerebral and cerebellar cortex, hypothalamus/thalamus, and hippocampus (Figure 9). Particular lesions presentation in the corresponding brain areas (i.e., cerebral and cerebellar cortex) likely reflect particular damaging effects (i.e., cerebellar cortex appears to be the most affected in the early course (Figure 9), cerebral cortex later less affected, microglial recruitment presented later). Illustrative is also the rapid worsening. With saline medication initiated at 15 min ligation-time, controls at 1 min thereafter showed scant karyopyknosis in cerebral cortex, with increasing number of karyopyknotic neurons at 5 min thereafter. Complete infarction in the controls appears at 24 h, and marked karyopyknosis at 48 h. In all BPC 157 rats only a few karyopyknotic neurons were found (Figure 11). That consistent neuroprotective effect appeared in all brain areas (Figure 12). Finally, the consistent less microglia presentation in the later period is likely along with the possible role of the microglia activation, and definitive beneficial effect that was achieved in BPC 157 rats (Figure 13). Additionally, using CD68 KP1, CD68 (PG-M1), and CD163 immunohistochemical antibodies for labeling microglial cells in rats euthanized at 48 h, results were consistent with semiquantitative microglial scoring performed on HE stained slides. The majority of cells corresponded to M1 macrophages, while only scattered M2 type macrophages were found in control rats In BPC 157 treated rats less than 5 microglial cells were found per 1 mm^2^, primarily M1 macrophages (Figure 14).

Furthermore, unlike BPC 157 rats, control rats exhibited significant lung, liver, kidney and heart lesions (Figure 10, Figure 15 and Figure 16). Specifically, there were the liver and lung congestion with intralveolar hemorrhage and pyknotic hepatocyte nuclei, marked congestion in heart tissue, within myocardium and large coronary branches, hyaline tubular cylinders, cell degeneration of proximal and distal tubule with cytoplasmic vacuolization in kidney after both in short-term (15 min ligation time, and period thereafter) and in long-term (24 h, 48 h). As emphasized, BPC 157 rats regularly did not show the liver and lung congestion, lung hemorrhage and hepatocyte damage or heart tissue congestion or kidney lesions. Finally, the ligated superior sagittal sinus rats present the erosive gastritis and duodenitis, which are however absent in BPC 157 treated rats. In the spleen, BPC 157-treated rats have less apparent sinusoidal congestion, and dilatation and enlargement of red pulp leading to reduction of white pulp.

In summary, the occluded superior sagittal sinus study evidences that BPC 157 therapy rapidly attenuates the brain swelling, rapidly eliminates the increased pressure in the ligated superior sagittal sinus, along with eliminating the severe portal and caval hypertension and aortal hypotension and emerging the rapid collateral vessels recruitment, both centrally and peripherally. The brain, heart, lung, liver, kidney, gastrointestinal lesions, thrombosis attenuation appears as the automated result.

## 4. Discussion

BPC 157 may consistently rescue the rats, which have permanent occlusion of the superior sagittal sinus, but drew up an apparent collapse of the otherwise imminent severe noxious course. Support implies all assessments used (i.e., gross recording, venography, ECG, pressure, microscopy and biochemistry assessment). Thus, it appears that BPC 157 therapy, may transmit a strong one beneficial effect, including the cerebral and cerebellar cortex, hypothalamus, thalamus, hippocampus, and extend the therapy success seen with the rescue of the rat peripheral venous occlusion syndromes [20,21,22,23,24,25,26,27].

This may be important since without therapy, the occluded superior sagittal sinus rapidly provoked a particular syndrome, the severe brain disturbances, brain swelling and lesions (i.e., confluent infarction encompassing most of the cerebral cortex) and highly increased positive pressure values and thrombosis in the superior sagittal sinus. In the periphery, as an instant progression from the central disturbance, a multiorgan failure syndrome appeared, comparable to the deadly syndromes induced by the peripheral vessels occlusion, the suprahepatic occlusion of the inferior caval vein (Budd–Chiari syndrome) or Pringle maneuver ischemia, reperfusion (i.e., heart, lung, liver, kidney, gastrointestinal lesions and thrombosis progression) [26,27]. Commonly, there were no presentation of the collateral pathways, centrally or peripherally. Thus, without therapy, a life-threatening course characterizes complex syndrome of the definitive superior sagittal sinus occlusion.

Most certainly, the sinus ligated to the bone ascertained definitive occlusion, skull bone overlying the sinus eliminated the risk of sinus tear. Thereby, such a ligation method of the superior sagittal sinus means permanent occlusion, permanent alteration of blood flow, and permanent increased pressure in the superior sagittal sinus. In this, calvariectomy and/or laparotomy (used to assess intracranial (superior sagittal sinus), portal, inferior caval vein and aortal pressure, and brain swelling and organs lesions and vessels presentation) since used in the therapy to counteract increased intracranial pressure and abdominal compartment syndrome [56,57], did not further contribute to the worst circumstances created by the occlusion of the superior sagittal sinus. Thereby, the evidenced high superior sagittal sinus, portal and caval hypertension and aortal hypotension are along with the rapid worsening that would appear along with the superior sagittal sinus occlusion. As before in the peripheral venous occlusion studies [20,21,22,23,24,25,26,27], clear prime injury site (i.e., occlusion by ligation), not removable, would continuously perpetuate injurious course, a point that can be hardly secured otherwise (i.e., FeCl3 thrombosis) [58].

Vice versa, confronted with the permanent central venous occlusion, the therapy effect is starting centrally, from the occluded superior sagittal sinus, as a prime noxious event (note, instead increased positive pressure, BPC 157-rats presented the negative pressure values close to those noted in the superior sagittal sinus of the healthy rats). Evidently, it successfully extended the previous BPC 157′s “bypassing key”, activated specific collateral pathways that successfully overwhelmed peripheral major veins occlusion and largely attenuated/eliminated the otherwise deadly peripheral venous occlusion syndromes, abrogated portal and caval hypertension and aortal hypotension [26,27]. As a continuation, we revealed the novel applicable therapy evidence, centrally, “bypassing key” activated specific collateral pathways, and BPC 157 effectiveness, µg-ng, regimens, given topically at the swollen brain, intraperitoneally or intragastrically, as an early or delayed application. The shunts are through ophthalmic vein, angularis vein, facial anterior and posterior vein, and facial vein, or through cerebri superior veins, sinus cavernosus, sinus petrosus superior and inferior, sinus transversus, through jugular external vein, subclavia vein through superior caval vein. Thereby, there was rapid attenuation of the brain swelling. The rapid elimination of the increased pressure in the ligated superior sagittal sinus, rapid elimination of the severe portal and caval hypertension and aortal hypotension, and rapid collateral vessels recruitment, abrogated venous and arterial thrombosis and recovery of the organs lesions, consistently occurred. Together, the beneficial action is going on both centrally and peripherally, overwhelming both central and peripheral harms of the permanent central venous occlusion. There is the activated azygos vein pathway, left superior caval vein-azygos vein-inferior caval vein, and, consequently, congested inferior caval vein and superior mesenteric vein became decongested reflecting elimination of the otherwise severe caval and portal hypertension. This occurred as it had been noted in Budd–Chiari syndrome BPC 157 therapy, with either intraperitoneal or per-oral application [27], indicative for the syndrome of the occluded superior sagittal sinus.

Therefore, presenting with the normalized pressures in the BPC 157-rats, abrogated organs lesions and counteracted thrombosis, these activated shunts represent the rapidly established alternative equilibrium, which is well functioning (i.e., consistently maintained undisturbed blood pressure in the superior sagittal sinus) in the compensation of the otherwise harmful conditions of the central venous occlusion. Even more, this particular vascular network organization allows it to work up to potential of further worsening, which may appear. Thus, there is the essential acting against additional intracranial hypertension, taken as the final disturbance, disclosed by the either of the central or peripheral intravenous challenges. It may be that BPC 157 therapy effect as general concept well covers particular circumstances since intracranial hypertension may differ from each other. The counteraction occurred against the severe additional intracranial hypertension (and against additionally increased portal and caval hypertension and aortal hypotension), centrally induced, since caused by additionally increased venous resistance/pressures, within the cranium (as shown by the highest venous hypertension additionally originated within the cranium, additional volume inside the sinus lumen). Likewise, counteraction occurred against the alternate additional intracranial hypertension (and against no changed portal and caval hypertension and aortal hypotension), a prime central target, caused in the periphery, within the neck (additional volume in the right external jugular vein) or in the abdomen (application in the inferior caval vein) [59]. If this beneficial effect and resolving adequately the anatomical imbalance in venous drainage would not occur, centrally, a harmful inability to drain venous blood adequately for a given cerebral blood inflow without raising venous pressures, suddenly goes to such venous and intracranial hypertension [60,61,62,63,64], as seen in the controls with the permanently occluded superior sagittal sinus. To this point may serve the BPC 157 counteraction of the microglia recruitment. Namely, microglia serves as resident immune cells of the central nervous system that constantly monitor the microenvironment of the neural parenchyma participating in synaptic maintenance [65,66]. Thereby, there is not only impact of BPC 157 on microglia recruitment nonspecifically, but macrophage polarization by activation of M1 macrophages, which induce prototypic inflammatory responses, can be reduced by the BPC 157.

Thus, less microglia recruitment in BPC 157 rats is important for the maintained neuroprotection, since microglia become activated as a response to neuronal damage, changing into an “ameboid” form that can migrate to sites of injury where they phagocytose debris and dying cells [65,66]. Activated microglia can also trigger inflammation signaling to neuroglia and circulating monocytes [65,66]. The recruitment of microglia in the vicinity injured neuronal cells and their subsequent activation soon after the neuronal injury occurs seems to be found long before the clinical onset of CNS injury and may contribute to both neuronal damage and pathogenesis [65,66]. Thus, this means rapid amelioration. As pointed out [60], it has to be the counteraction of the reduced capillary perfusion pressure, increased cerebral blood volume, venous flow obstruction, increased intracranial pressure and blood-brain barrier disruption, resulting in a decreased cerebral blood flow [60]. Thereby, as mentioned, it may be relevant the described immediate gross disappearance of the swollen brain presentation, and microscopically only a few karyopyknotic neurons upon administration of the BPC 157 regimens, given locally at the swollen brain, or intragastrically or intraperitoneally, thus, evidencing a consistent BPC 157 effect, as shown by the direct recording. The attenuated were progressive cerebral edema, which may the net capillary filtration increase, and intracerebral and subarachnoid hemorrhage, which may additionally compromise the brain tissue [60]. Interestingly, it may be that these findings are showing that the minimal effective drug concentration very rapidly reaches the brain (i.e., at 1 min), because there is no change in time and dose-concentration. On the other hand, in particular considering BPC 157 intragastric application, these findings are along with the abovementioned BPC 157 supposed conceptual significance as an original cytoprotective anti-ulcer peptide resistant to human gastric juice [5,6,7,8,9,10,11,12,13,14,15,16,17,18,19], and essential part of function of brain–gut axis and gut–brain axis.

Simultaneously, this includes, as described before [20,26,27,67], even with delayed therapy application, no thrombosis (or at least marked thrombosis attenuation) as a point of the eliminated/attenuated stasis as well as resolved Virchow triad situation [20,21,22,23,24,25,26,27]. Otherwise, without therapy, reflecting a general stasis (i.e., large volume trapped in the CNS and portal and caval vein tributaries (i.e., seen with inferior caval vein venography) may perpetuate the brain and heart ischemia as well), thrombosis rapidly appears (i.e., 15 min). It was noted in all vessels tested, venous (centrally (i.e., superior sagittal sinus); peripherally (i.e., portal, superior mesenteric, lienal and inferior caval vein)) and arterial (i.e., abdominal aorta thrombosis, thus low-flow state caused by cardiac dysfunction or sever volume depletion [68]). These rats with severe brain injuries exhibited prolonged QT-interval, subendocardial congestion, heart dilatation and insufficient heart (contrast) filling, severe capillary congestion in alveolar septa, edematous exudate in alveoli) (heart and lung as additional prime targets). Then, they consequently exhibited the liver failure (substantial congestion of central vein as well as branches of portal veins in portal triads), gastric and duodenal hemorrhagic lesion, due to prominent portal and caval hypertension, as noted in the peripheral vein occlusion syndromes. Thus, as cause consequence relations, evidenced also in the previous venous occlusion studies, that course mimics supporting associations noted also in patients. Illustrative may be the venous infarct or edema of frontal lobes in anterior skull base lesions where the anterior superior sagittal sinus is ligated and transected [1]; subarachnoid hemorrhage, intracranial hypertension [69,70,71] and subarachnoid hemorrhage and the QTc duration normalization as the patients’ neurological status improved [72]. Consequently, with BPC 157 therapy, the clear counteraction, which occurred in the rats which have the permanent occlusion of the superior sagittal sinus, as before in the Budd–Chiari syndrome-rats or Pringle maneuver ischemia, reperfusion-rats [26,27] and other venous occlusion studies [20,21,22,23,24,25,26,27], may suggest possible BPC 157 therapy effect in patients as well. Contributory factors in addition to, or likely due to the described effect on vessel recruitment [20,21,22,23,24,25,26,27], may be that BPC 157 counteracts in rats various encephalopathies [33,34,35,36,37,38,39,40,41]. Likely in the same way, along with encephalopathies [33,34,35,36,37,38,39,40,41], BPC 157 counteracts multiple pathology in the gastrointestinal tract and liver [33,34,35,36,37,38,39,40,41], and various arrhythmias [73,74,75,76,77] (in particular, BPC 157 therapy normalizes the QTc duration in rats treated with neuroleptics and prevents and recovers chronic heart failure [75,76]).

In conclusion, the deciding result is the exemplified principle [7,8], the resolved cytoprotective equation, endothelium maintenance → epithelium maintenance = blood vessel recruitment and activation towards defect or bypassing vessel occlusion [20,21,22,23,24,25,26,27], also in the central venous occlusion. With pentadecapeptide BPC 157 application, rapid elimination of the increased pressure in the ligated superior sagittal sinus, the eliminated severe portal and caval hypertension and aortal hypotension, and rapid collateral vessels recruitment, counteracted multiorgan failure syndrome denotes the essential proof-of-principle. Further specificity may be the noted special interaction with NO-system and NO-agents in various models and species [7,8,14] since BPC 157 induced the NO-release of its own [78,79]. The effects on blood pressure and thrombocytes function maintenance [78,80,81] are suggestive, BPC 157 may counteract induced hypertension and pro-thrombotic effect (L-NAME), and induced hypotension [78,81] and anti-thrombotic (L-arginine) effect [81], and finally, exert a vasomotor tone carried out through BPC 157 specific activation of Src-Caveolin-1-endothelial nitric oxide synthase (eNOS) pathway [47]. Thus, the collected evidence can guarantee the particular background, which still needs further elaboration. The evidence is that BPC 157 exhibited a specific effect on the *Egr, Nos, Srf, Vegfr, Akt1, Plc**ɣ,* and *Kras* pathways in the vessel which provides an alternative operating pathway (i.e., left ovarian vein as the key for the infrarenal occlusion-induced inferior caval vein syndrome in rats) [20]. In hippocampal tissues, mRNA expression studies at 1 h and 24 h, and strongly elevated (*Egr1, Akt1, Kras, Src, Foxo, Srf, Vegfr2, Nos3, Nos1*) and decreased (*Nos2, Nfkb*) gene expression (*Mapk1* not activated) also support BPC 157 beneficial effect on brain lesions, given in reperfusion in stroke-rats [30]. BPC 157 therapy counteracted both early and delayed neural hippocampal damage. Full functional recovery can restore recognition memory deficits along with the therapy effect [30]. This may be an indicator of how BPC 157 may act [30]. To this point, note that the consistently effective used range of BPC 157 (µg-ng) application and used regimens (at the swollen brain, intraperitoneally or intragastrically) may support each other’s effects. This may be a physiological role (in situ hybridization and immunostaining BPC 157 in human gastrointestinal mucosa, lung bronchial epithelium, epidermal layer of the skin and kidney glomeruli), and BPC 157 may act as stabilizer of cellular junction, leading to significantly mitigated leaky gut syndrome, via increasing tight junction protein ZO-1 expression, and transepithelial resistance [19]. Likewise, the mRNA of inflammatory mediators (iNOS, IL-6, IFNγ and TNF-α) are inhibited, with increased expression of HSP 70 and 90, and antioxidant proteins, such as HO-1, NQO-1, glutathione reductase, glutathione peroxidase 2 and GST-pi [19].

Finally, this study should overwhelm the general point that animal studies per se may be cautious regarding their results, and the relative paucity of the BPC 157 clinical data [7,16]. Namely, it should be noted that BPC 157 was proved to be efficacious in the ulcerative colitis, both in clinical settings [82,83] as well as in the experimental rats ischemic/reperfusion vascular ulcerative colitis studies [21] and other ulcerative colitis models [10,11,19,33,34,35,36,37,38,39,40,41]. A particular point is a very safe profile (LD1 could be not achieved) [7,16], a point recently confirmed in a large study of the Xu and collaborators [83]. In this context, the role of the animal model is indispensable, and the practical indicative evidence is even more important. Thus, BPC 157 therapy rapidly overwhelms the occluded superior sagittal sinus in rat. This also emphasizes the bypassing principle in the central venous occlusion. Rapidly activating alternative bypassing pathways override the permanent ligation of the superior sagittal sinus and limitations which otherwise make the study of the therapeutic strategies difficult.

## Figures and Tables

**Figure 1 biomedicines-09-00744-f001:**
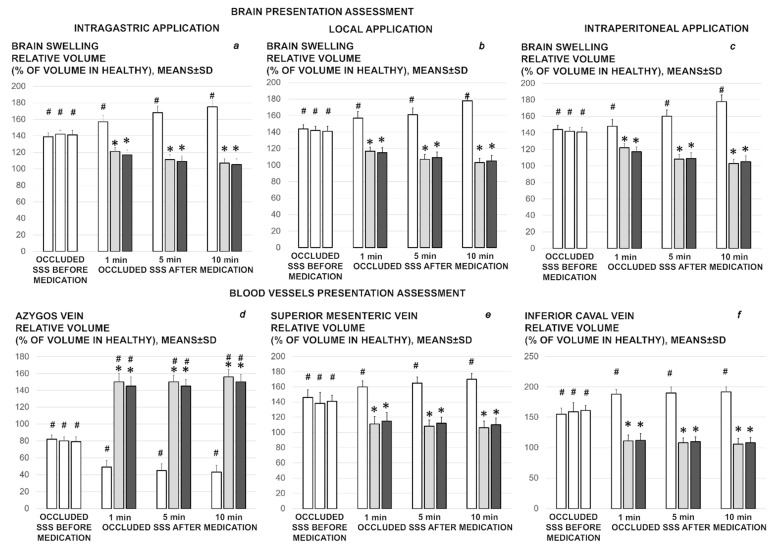
Brain and blood vessels presentation assessed as the relative volume, % of volume in healthy, before ligation (100%), and then in the rats with occluded superior sagittal sinus, immediately after ligation (occluded superior sagittal sinus (SSS) before medication, white bars), and then after therapy (occluded superior sagittal sinus (SSS) after medication) at 1 min, 5 min, and 10 min ligation-time. Therapy was saline 5 mL/kg (white bars), BPC 157 10 µg/kg (light gray bars), 10 ng/kg (dark gray bars). In the assessment of the brain swelling progress or elimination, the medication effect was shown when given intragastrically (**a**), locally at the swollen brain (**b**), or intraperitoneally (**c**); or intragastrically with azygos (**d**), superior mesenteric (**e**), and inferior caval vein (**f**) assessment (while the effects of the intraperitoneal and local application regimens, providing the similar results were not specifically shown). Means ± SD, * *p* ˂ 0.05, at least vs. corresponding control; ^#^ *p* ˂ 0.05, at least vs. regular healthy values.

**Figure 2 biomedicines-09-00744-f002:**
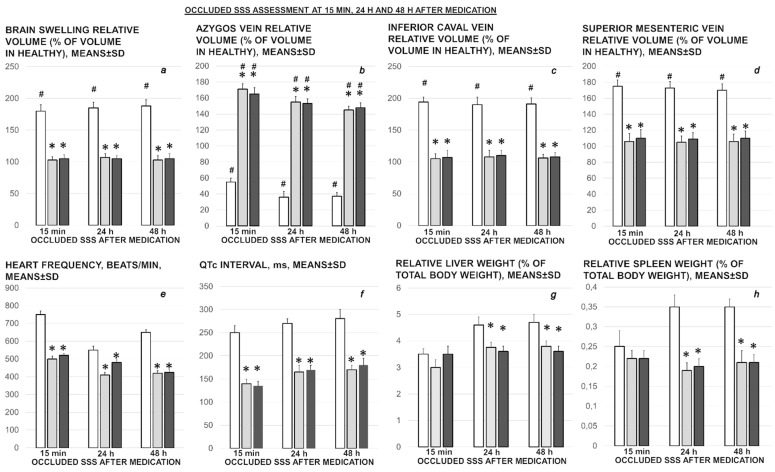
Brain (**a**) and blood vessels (azygos vein (**b**), inferior caval vein (**c**), superior mesenteric vein (**d**)) presentation assessed as the relative volume, % of volume in healthy, before ligation (100%), and then in the rats with occluded superior sagittal sinus (SSS), after therapy at 15 min, 24 h, and 48 h ligation-time. Heart frequency, beats/min (**e**), and QTc interval, ms (**f**), relative liver (**g**) and spleen (**h**) weight (% of total body weight) (low, right) at 15 min, 24 h, and 48 h ligation-time. Therapy was saline 5 mL/kg (white bars), BPC 157 10 µg/kg (light gray bars), 10 ng/kg (dark gray bars) given intragastrically at 1 min ligation time while the effects of the intraperitoneal and local application regimens, providing the similar results were not specifically shown. Means ± SD, * *p* ˂ 0.05, at least vs. corresponding control; ^#^ *p* ˂ 0.05, at least vs. regular healthy values (100%).

**Figure 3 biomedicines-09-00744-f003:**
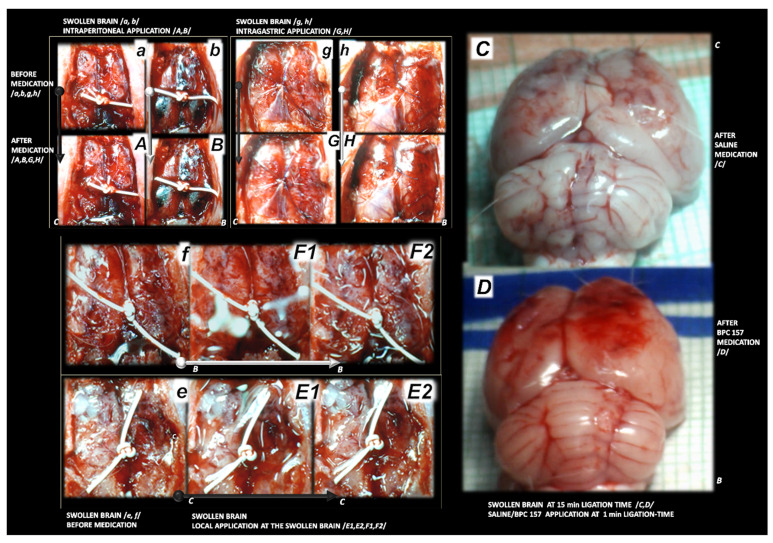
Rat brain swelling presentation at 15 min ligation time in rats with superior sagittal sinus ligation. Rats received saline (**C**) or BPC 157 (**B**) and indicated with small white C and small white B close to the corresponding presentation. Swollen brain before medication (**a**,**b**,**e**–**h**), and after medication, given earlier at 1 min ligation-time (right, (**C**,**D**)). Therapy applied at 15 min ligation-time (**A**,**B**,**E1**,**E2**,**F1**,**F2**,**G**,**H**), presented the effect of medication at 5 min therapy-time (medication was given intraperitoneally (**A**,**B**) (upper, left, saline (small (**C**)), BPC 157 (small (**B**)), intragastrically (**G**,**H**) (upper, middle, saline (small (**C**)), BPC 157 (small (**B**)). Local application at the swollen brain. Swollen brain before medication (**e**,**f**). For medication given locally at the swollen brain (arrows), 30 s therapy-time (brain under solution, small C (control, saline), small B (BPC 157 solution)) and 45 s therapy-time (immediately upon solution resorption, small C (control, saline), small B (BPC 157 solution)) (low, (**E1**,**E2**) (saline), (**F1**,**F2**) (BPC 157)). Aggravation of the swelling, saline (intraperitoneally) (**A**), intragastrically (**G**); saline (topically) (**E1**) (30 s therapy-time), (**E2**) immediately upon solution resorption (45 s therapy-time). Attenuated swelling, BPC 157 (intraperitoneally (**B**); intragastrically (**H**) (at 5 min therapy-time); BPC 157 (topically) (**F1**) (30 s therapy-time), (**F2**) immediately upon solution resorption (45 s therapy-time). (**C**,**D**). Characteristic gross brain presentation with medication saline (**C**) or BPC 157 (**D**) given intraperitoneally at 1 min ligation time.

**Figure 4 biomedicines-09-00744-f004:**
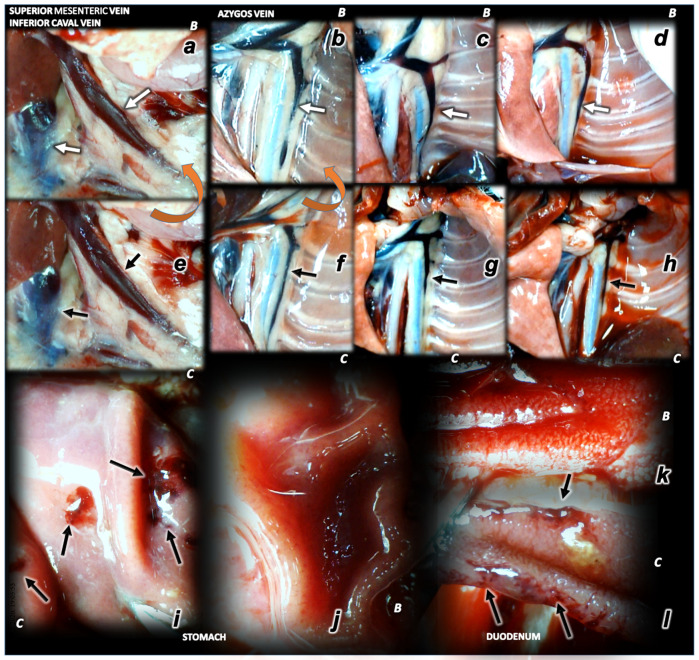
Illustrative presentation of blood vessels (superior mesenteric vein, inferior caval vein, azygos vein) (**a**–**h**), stomach and duodenal lesions (**i**–**l**). Rats received saline (**C**) or BPC 157 (**B**). Superior mesenteric vein, inferior caval vein (**a**,**e**), azygos vein (**b**–**d**,**f**,**g**). Upon superior sagittal sinus ligation, as illustration of the severe portal and caval hypertension, and not functioning connective pathways, congested superior mesenteric and inferior caval vein (arrows) (**e**), collapsed azygos vein (arrows) (**f**). Azygos vein remains weakly presented at 24 h (azygos vein, low, middle, (**g**)) and 48 h ligation time (azygos vein, low, right, (**h**)). Upon BPC 157 intragastric application, superior mesenteric and inferior caval veins are rapidly decongested (**a**) and azygos vein reopen (arrows) (upper, left, (**b**)) sustainably presented also at 24 h (azygos vein, upper, middle (**c**)) and 48 h ligation time (azygos vein, upper, right (**d**)) (the effect along with elimination of the portal/caval hypertension, and elimination of the intracranial hypertension). Low. At 15 min ligation time, hemorrhagic lesions in the duodenum (right (**l**)) and stomach (left (**j**)) of the control rats (arrows), and spared mucosa in the duodenum (**k**) and stomach (**i**) of the BPC 157 treated rats. Magnification ×30.

**Figure 5 biomedicines-09-00744-f005:**
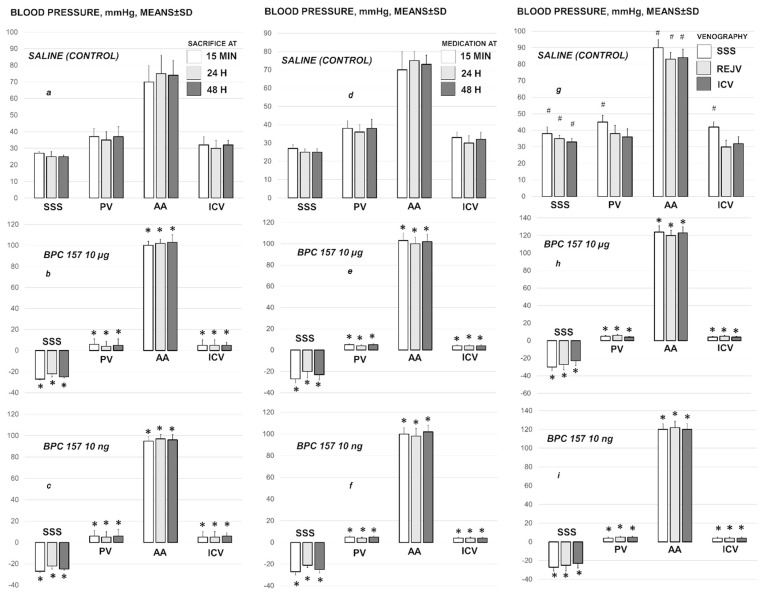
Blood pressure of the rats with ligated superior sagittal sinus in the superior sagittal sinus (SSS), portal vein (PV), abdominal aorta (AA) and inferior caval vein (ICV), at the end of the 5 min assessment following medication (BPC 157 10 µg/kg, 10 ng/kg; saline 5 mL/kg) given intraperitoneally. Early medication at 1 min ligation time, blood pressure assessment before euthanized at 15 min (**a**), 24 h (**b**) and 48 h (**c**) ligation time. Delayed medication at 15 min (**d**), 24 h (**e**) and 48 h (**f**) ligation time. (**g**–**i**) For verification of the effect of venography, assessment at 5 min upon the additional 1 mL of saline application in the superior sagittal sinus (SSS), right external jugular vein (REJV) or inferior caval vein (ICV), inducing venous hypertension additionally originated within the cranium, inside the sinus lumen, neck or abdomen, at 15 min, and subsequent immediate ip application of saline or BPC 157. Similar effects were obtained with BPC 157 intragastric or local application at the swollen brain (data not specifically shown). Means ± SD, * *p* ˂ 0.05, at least vs. control; ^#^
*p* ˂ 0.05, at least vs. regular saline-control in non-venography studies.

**Figure 6 biomedicines-09-00744-f006:**
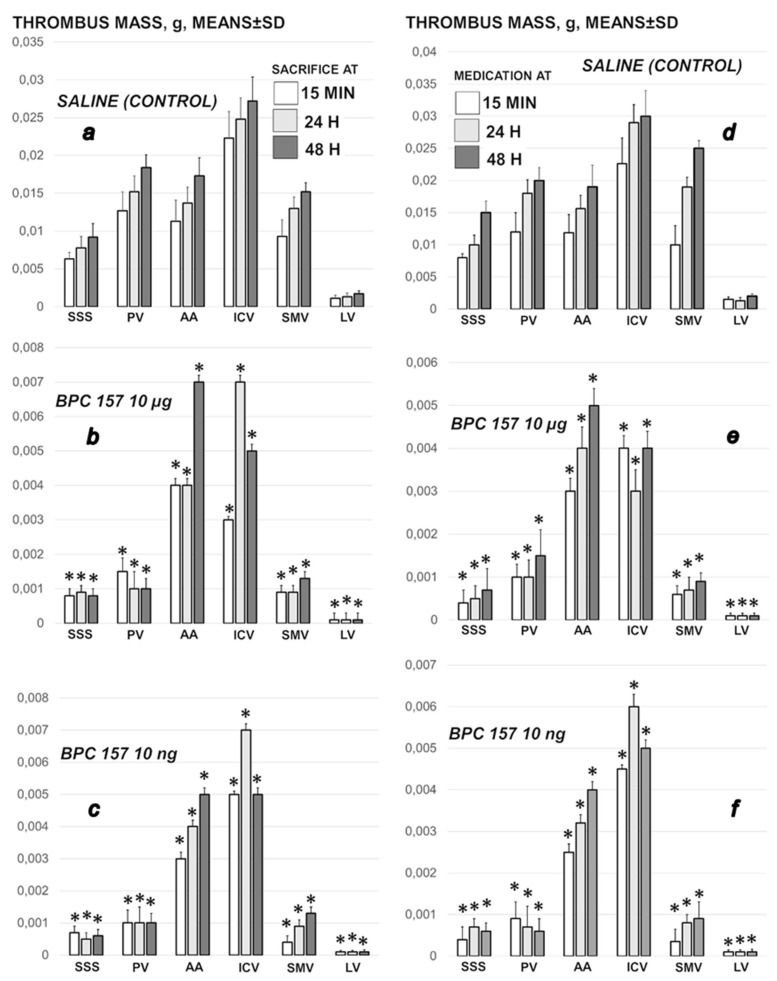
Thrombus presentation in the rats with ligated superior sagittal sinus in the superior sagittal sinus (SSS), portal vein (PV), abdominal aorta (AA), inferior caval vein (ICV), superior mesenteric vein (SMV) and lienal vein (LV) at the end of the 5 min assessment following medication (BPC 157 10 µg/kg, 10 ng/kg; saline 5 mL/kg) given intraperitoneally. **Left:** Early medication at 1 min ligation time, blood pressure assessment before sacrifice at 15 min, 24 h and 48 h ligation time (**a**–**c**). **Right:** Delayed medication at 15 min, 24 h and 48 h ligation time (**d**–**f**). Similar effects were obtained with BPC 157 intragastric or local application at the swollen brain (data not specifically shown). Means ± SD, * *p* ˂ 0.05, at least vs. control.

**Figure 7 biomedicines-09-00744-f007:**
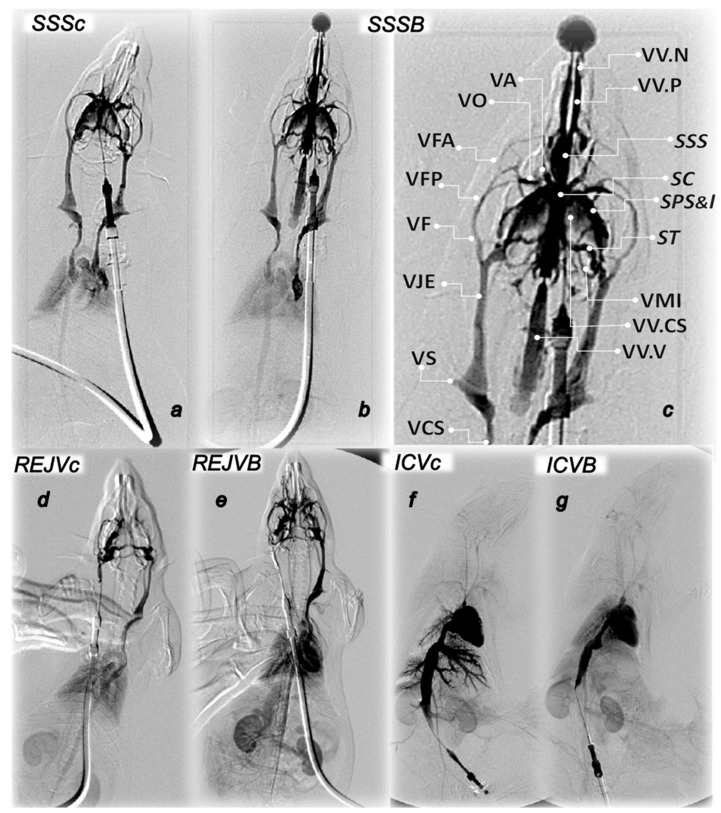
Along with saline (**c**) or BPC 167 (**B**) presentation of venography in superior sagittal sinus (SSS) (**a**–**c**), right external jugular vein (REJV) (**d**,**e**) and inferior caval vein (ICV) (**f**,**g**). SSS. **Upper Left:** Control venography in superior sagittal sinus (*SSSc*) (**a**). Retrograde filling of both jugular external veins and both superior caval veins. Pulmonary congestion with heart dilatation and without inflow of contrast media in the heart. Retrograde filling of pale inferior caval vein. **Upper Right:** BPC 157 venography in superior sagittal sinus (*SSSB*) (**b**,**c**). Retrograde filling of both jugular external veins and both superior caval veins with vertebral veins. There is no sign of pulmonary congestion and heart dilatation and with normal filling of contrast media in the heart. Retrograde filling of wider inferior caval vein with inflow of hepatic veins, both renal veins with parenchymal phase visualization of both kidneys and adrenals. Presentation of the pterigopalatinal veins (VVP) and nasal veins (VV.N) and presentation of the ophthalmic vein (VO), angularis vein (VA), facial anterior and posterior vein (VFA, VFP), and facial vein (VF), or through cerebri superior veins (VV.CS), sinus cavernosus (SC), sinus petrosus superior and inferior (SPS&I), sinus transversus (ST), through jugular external vein (VJE), subclavia vein (VS) through superior caval veins (VCS). **Lower Left:** REJV. Control venography through right external jugular vein (*REJVc*) with retrograde filling of left external jugular vein and left superior caval vein (left, *REJVc*) (**d**). There is pulmonary congestion, with insufficient heart contrast filling. Pale visualization of aortic arch and ascendent aorta, without supra-aortic branches. Retrograde filling of inferior caval vein without visualization of renal veins with pale parenchymal phase of both kidneys and adrenals. BPC 157 venography through right external jugular vein (*REJVB*) (**e**) with retrograde filling of left external jugular vein and left superior caval vein. Partial visualization of azygos vein. There is no sign of pulmonary congestion, with normal heart contrast filling. Significant visualization of aortic arch, with supra-aortic branches-both common carotid arteries. Clear visualization of thoracic and abdominal aorta. Retrograde filling of dilated inferior caval vein with paravertebral venous plexus. Slight visualization of hepatic veins inflow. Visualization of both renal veins with strong parenchymal phase of both kidneys and adrenals. In addition, there is visualization of portomesenteric veins and intestinal branches. **Lower Right:** ICV. Control venography through inferior caval vein (*ICVc*) (**f**) shows huge congestion within prominent hepatic veins without parenchymal liver phase. In addition, there is severe congestion in pulmonar artery ramification, without parenchymal lung phase and dilatation of the heart with filling of contrast media. There is also renal veins congestion. In the same phase opacification of thin aortic lumen and iliac arteries with pale opacification of the superior mesenteric artery. BPC 157 venography through inferior caval vein (*ICVB*) (**g**) goes without congestion within hepatic veins with parenchymal liver phase. In addition, there is mild congestion in pulmonary artery with parenchymal lung phase, normal heart filling with contrast media. Visualization of both kidneys in parenchymal kidney phase. In the same phase normal aortic lumen with prominent opacification of the superior mesenteric artery and intestinal.

**Figure 8 biomedicines-09-00744-f008:**
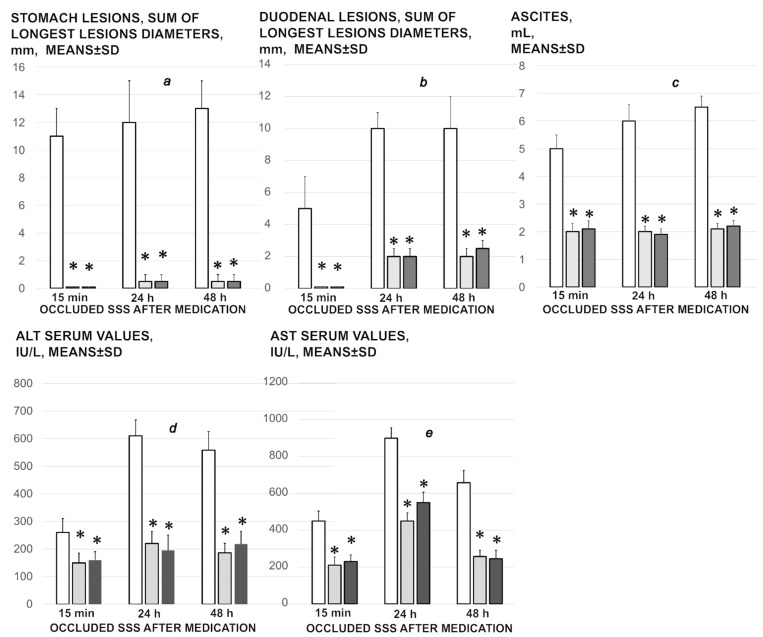
Stomach (**a**) and duodenal (**b**) lesions, ascites (**c**), ALT (**d**) and AST (**e**) serum values presentation in the rats with occluded superior sagittal sinus, after therapy at 15 min, 24 h, and 48 h ligation-time. Therapy was saline 5 mL/kg (white bars), BPC 157 10 µg/kg (light gray bars), 10 ng/kg (dark gray bars) given intragastrically at 1 min ligation-time while the effects of the intraperitoneal and local application regimen, providing the similar results were not specifically shown. Means ± SD, * *p* ˂ 0.05, at least vs. corresponding control.

**Figure 9 biomedicines-09-00744-f009:**
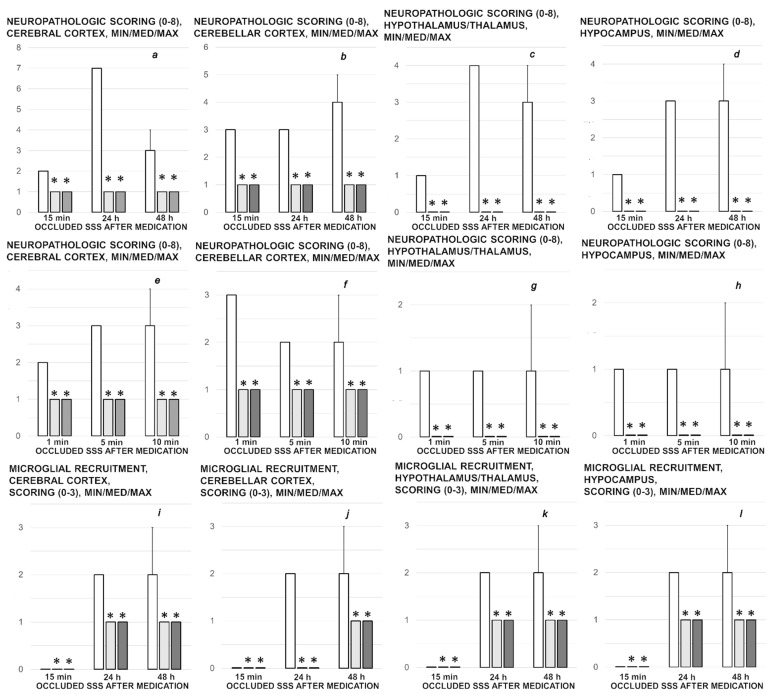
Neuropathologic scoring (necrosis + karyopiknosis) (0–8) (**a**–**h**), and microglial recruitment scoring (0–3) (**i**–**l**) in cerebral and cerebellar cortex, hypothalamus/thalamus and hippocampus after therapy (saline 5 mL/kg (white bars), BPC 157 10 µg/kg (light gray bars), 10 ng/kg (dark gray bars) given intragastrically at 1 min ligation time). Neuropathologic scoring (necrosis + karyopiknosis) (0–8) (**a**–**h**). Assessment carried out at 15 min, 24 h, and 48 h ligation-time (**a**–**d**) and at 1 min, 5 min and 10 min therapy-time (**e–h**). Microglial recruitment scoring (0–3) (**i**–**l**). Assessment at 15 min, 24 h, and 48 h ligation-time. Microglial recruitment was not present at 1 min, 5 min and 10 min therapy-time (data not specifically shown). The effects of the intraperitoneal and local application regimen, providing the similar results were not specifically shown. MIN/MED/MAX, * *p* ˂ 0.05, at least vs. corresponding control.

**Figure 10 biomedicines-09-00744-f010:**
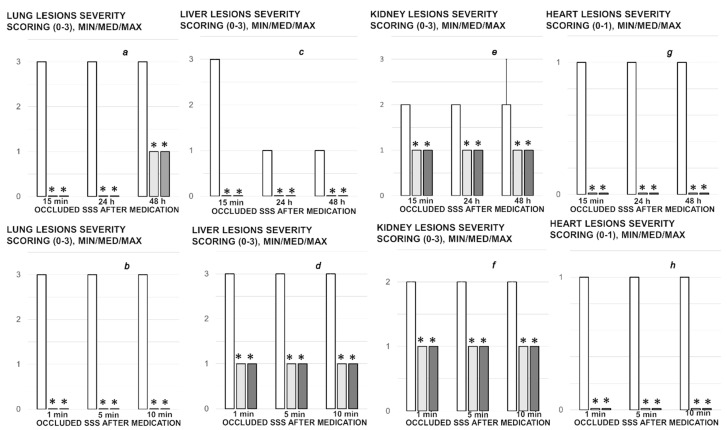
Organs lesions microscopy scoring. Lung (**a**,**b**), liver (**c**,**d**), kidney (**e**,**f**) (scored 0–3), heart (**g**,**h**) (scored 0–1) lesions. Assessment at 15 min, 24 h, and 48 h ligation-time (**a**,**c**,**e**,**g**). Assessment at 1 min, 5 min and 10 min therapy-time (**b**,**d**,**f**,**g**). Therapy was saline 5 mL/kg (white bars), BPC 157 10 µg/kg (light gray bars), 10 ng/kg (dark gray bars) given intragastrically at 1 min ligation-time. The effects of the intraperitoneal and local application regimen, providing the similar results, were not specifically shown. MIN/MED/MAX, * *p* ˂ 0.05, at least vs. corresponding control.

**Figure 11 biomedicines-09-00744-f011:**
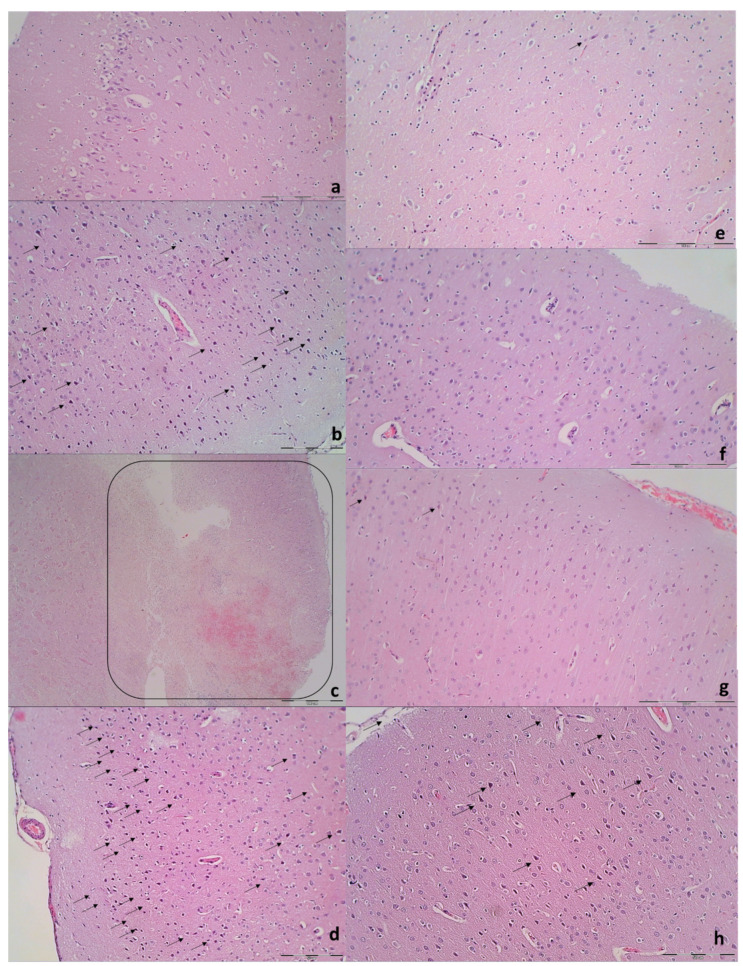
Neuropathologic changes in cerebral cortex areas (HE staining). Increased edema and congestion in control group (**a**–**d**) unlike BPC 157 rats (**e**–**h**). Application at ligation-time 15 min. (**a**) Control (1 min upon saline administration), a scant karyopyknosis in cerebral cortex, ligation-time 16 min; (**b**) Control (5 min upon saline administration), larger number of karyopyknotic neurons (arrows), ligation-time 20 min; (**c**) Control 24 h ligation-time (saline administration at 1 min ligation-time), complete infarction (rectangular marked area); (**d**) Control 48 h, ligation-time (saline administration at 1 min ligation-time, marked karyopyknosis (arrows)). BPC 157 rats exhibit a few karyopyknotic neurons (arrows) (**e**–**h**). (**e**,**f**) BPC 157 application at 15 min ligation-time, intragastric BPC 157 therapy, 1 min after application (**e**), or 5 min after application (**f**). (**g**,**h**) BPC 157 intragastric administration at 1 min ligation-time: 24 h ligation-time (**g**), 48 h ligation-time (**h**). Magnification ×200 ((**a**,**c**,**e**–**g**); scale bar 100 µm), ×200 ((**b**,**d**,**h**); scale bar 50 µm).

**Figure 12 biomedicines-09-00744-f012:**
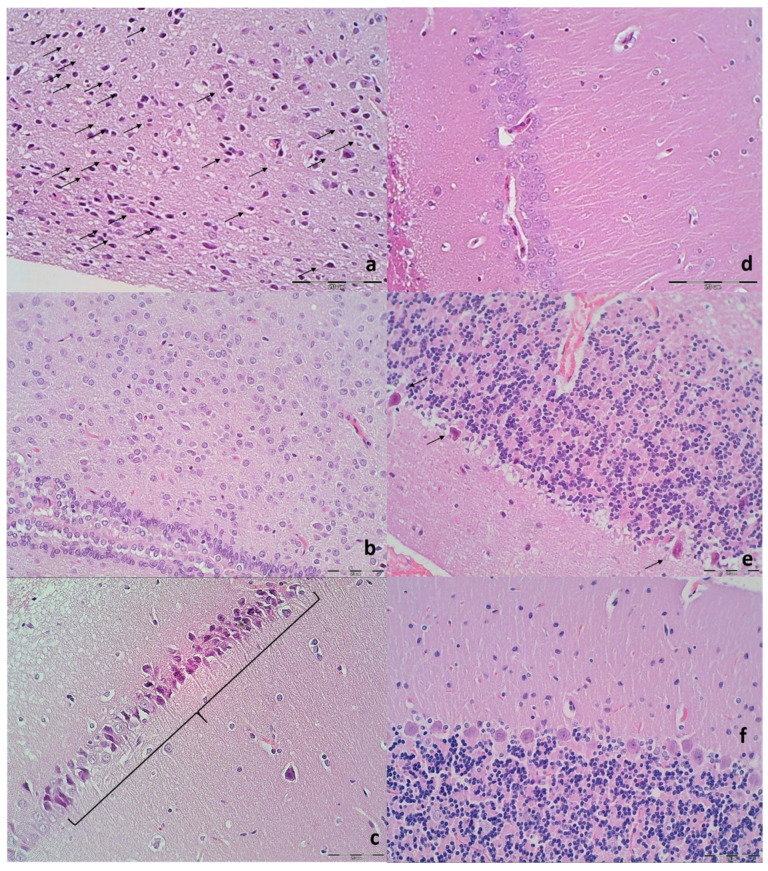
Illustrative neuropathologic changes in the rat cerebellar cortex, hypothalamus and hippocampus (HE staining). (**a**) Control showing marked karyopyknosis of hypothalamic neurons (arrows); (**b**) normal structure of hypothalamus in BPC 157 rats; (**c**) marked karyopyknosis of pyramidal cells of the hippocampus in control (marked with brace); (**d**) normal hippocampus in BPC 157 rats; (**e**) marked karyopyknosis, degeneration (arrows) and loss of Purkinje cells of the cerebellar cortex in control; (**f**) normal structure of cortex in BPC 157 rats. Magnification ×200 ((**a**–**f**); scale bar 20 µm).

**Figure 13 biomedicines-09-00744-f013:**
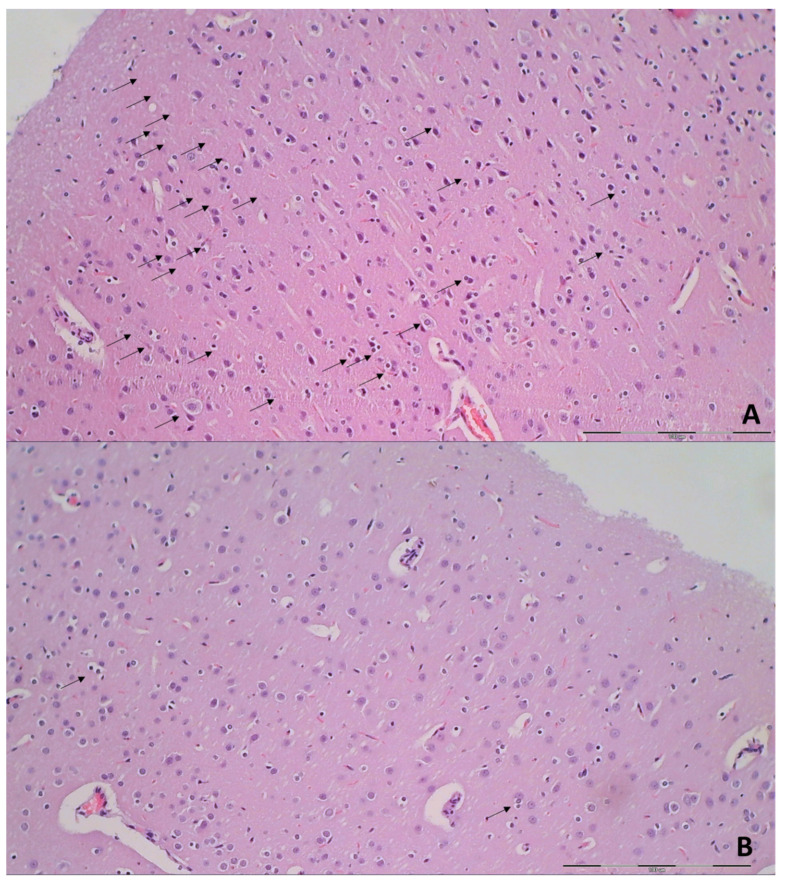
Microglia recruitment in cerebral cortex (HE staining). (**A**) Control groups after 24 h showed significantly more microglia cells with an amoeboid morphology in gray matter (arrows) (score 2). (**B**) BPC 157 after 24 h with less than 5 cells (arrows) (score 1). Magnification ×200 ((**A**,**B**); scale bar 100 µm; Olympus BX51 objective 20×; field area 1 mm^2^).

**Figure 14 biomedicines-09-00744-f014:**
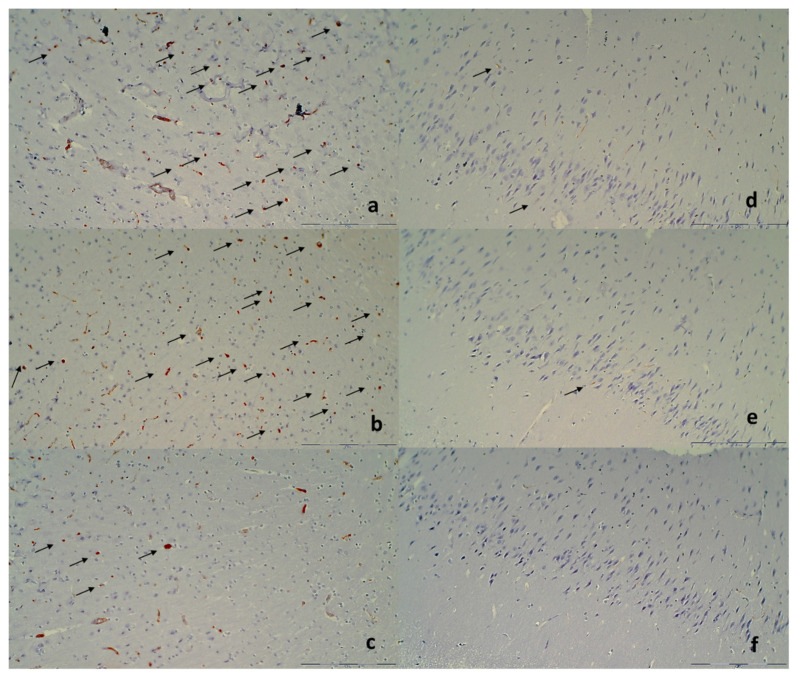
Immunohistochemical staining of microglial cells (**a**,**b**) CD68 KP1, (**c**,**d**) CD68 (PG-M1), (**e**,**f**) CD163 (marked with arrows; magnification ×400). (**a**) Control group after 48 h showed increased number of CD68 KP1 positive microglia cells with an amoeboid morphology in gray matter (score 3), while (**b**) BPC 157 with less than 5 cells (score 1). (**c**) Control group after 48 h showed increased number of CD68 (PG-M1) positive M1 type microglia cells with an amoeboid morphology in gray matter (score 3), while (**d**) BPC 157 with less than 5 cells (score 1). (**e**) Control group after 48 h showed less than 5 CD163 M2 type microglia cells with an amoeboid morphology in gray matter (score 1), while (**d**) BPC 157 with no cells (score 1). Magnification ×200 (scale bar 100 µm; Olympus BX51 objective 20×; field area 1 mm^2^; Note: endothelial cells were immunohistochemically labeled with all three antibodies but were excluded from scoring).

**Figure 15 biomedicines-09-00744-f015:**
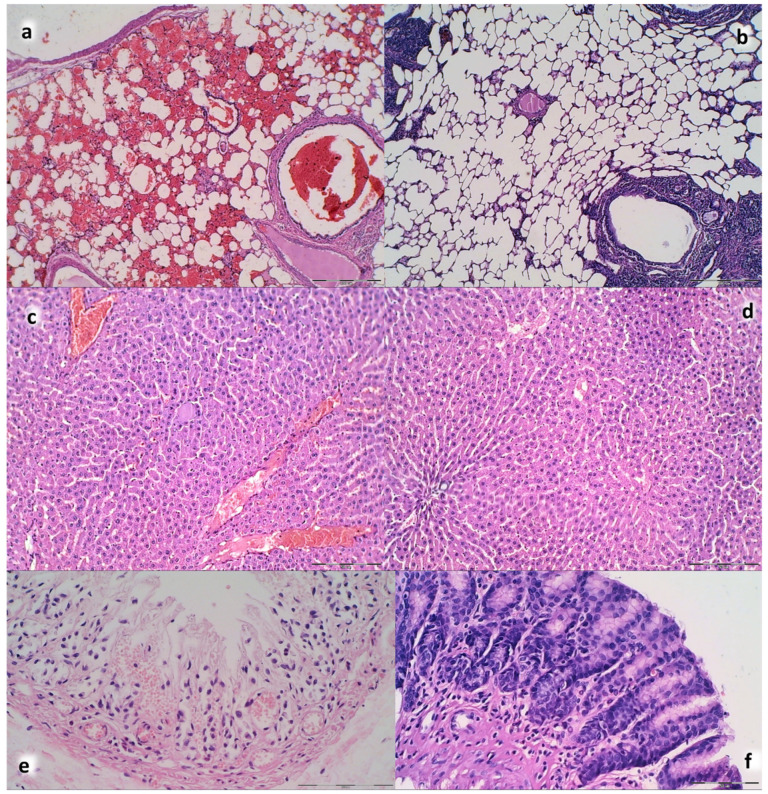
Illustrative microscopic presentation of the lung (**a**) (control), (**b**) (BPC 157)), liver (**c**) (control), (**d**) (BPC 157)), and stomach (**e**) (control), (**f**) (BPC 157)) (HE staining. (**a**–**d**) Presentation at the 20 min ligation-time, 5 min after saline (**a**,**c**) or BPC 157 (**b**,**d**) intragastric application. (**a**) Control rats with congestion and intralveolar hemorrhage; (**b**) no congestion and lung hemorrhage in BPC 157 rats; (**c**) liver parenchyma showed congestion in control; (**d**) BPC 157 rats showed no changes in liver parenchyma; (**e**,**f**) presentation at the 24 h ligation-time, saline or BPC 157 at 1 min ligation-time; (**e**) controls with erosive gastritis; (**f**) BPC 157 rats with no pathological changes. Magnification ×100 ((**a**,**b**) scale bar 100 µm), ×100 ((**c**,**d**) scale bar 50 µm), ×200 ((**e**) scale bar 100 µm), ×200 ((**f**) scale bar 50 µm).

**Figure 16 biomedicines-09-00744-f016:**
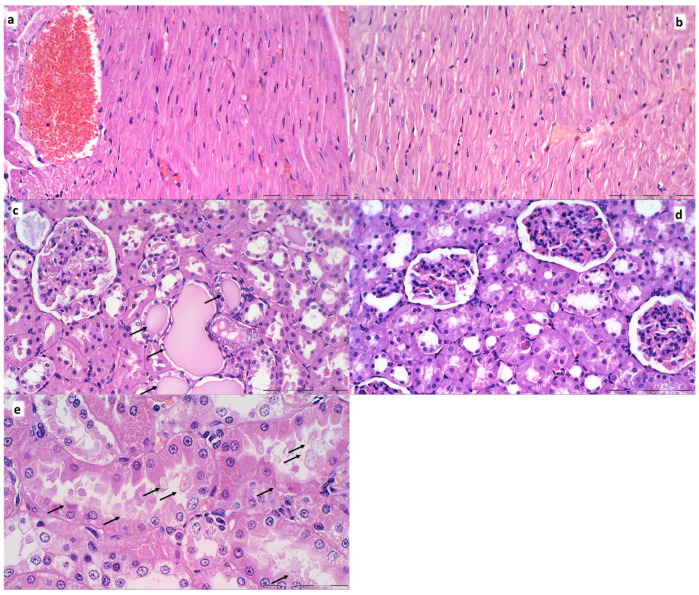
Illustrative microscopic presentation of the heart (**a**) (control), (**b**) (BPC 157)) and kidney (**c**,**e**) (control), (**d**) (BPC 157), (HE staining). Presentation at the 20 min ligation-time, 5 min after saline (**a**,**c**) or BPC 157 (**b**,**d**) intragastric application. Heart. (**a**) Control rats with marked congestion in the heart tissue, within myocardium and large coronary branches; (**b**) no vascular congestion in BPC 157 rats. Kidney. (**c**,**e**). Hyaline tubular cylinders (**c**), cell degeneration of proximal and distal tubule with cytoplasmic vacuolization in controls (**e**); (**d**) BPC 157 rats showed no pathological changes. Magnification ×200 ((**a**–**d**) scale bar 100 µm); ×400 ((**e**) scale bar 50 µm).

**Table 1 biomedicines-09-00744-t001:** The neuropathologic scores.

Brain Area	Grading	Percent Area Affected	Morphological Changes
Cerebral and cerebellar cortex, hypothalamus, thalamus, hippocampus	1	≤10	Small, patchy, complete or incomplete infarcts
	2	20–30	Partly confluent complete or incomplete infarcts
	3	40–60	Large confluent compete infarcts
	4	>75	In cortex; total disintegration of the tissue, in hypothalamus, thalamus, hippocampus; large complete infarcts
Cerebral and cerebellar cortex, hypothalamus, thalamus, hippocampus	1	≤20	A few karyopyknotic of neuronal cells
	2	50	Patchy areas of karyopyknotic areas
	3	75	More extensive of karyopyknotic areas
	4	100	Complete infarction

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
