# Peer review of "BPC 157 Therapy and the Permanent Occlusion of the Superior Sagittal Sinus in Rat: Vascular Recruitment"

_biomedicines, 2021, doi:10.3390/biomedicines9070744_

Round 1

Reviewer 1 Report

In this study, Gojkovic et al. presented the vascular recruitment and the effects of pentadecapeptide BPC 157 therapy and the permanent occlusion of the superior sagittal sinus in rat. Although the study is very elaborate and would be useful for researchers in vascular physiology area, the presentation of the data could be dramatically improved. There is a need for sham group. Also, the authors can increase the impact of their study by introducing another model of sagittal sinus thrombosis that is less invasive. Proposed changes are listed in the following section.

Major:

  1. Although the detail oriented study is well appreciated, the proposed publication is too long for smooth reading. I would suggest that the authors condense the introduction and discussion to emphasize the priority of the study. Also, the results should be rephrased to point to the relevant figure. Methods section can be compressed, and details can be submitted as the supplementary material. These changes will increase the reading flow.
  2. Data presentation should be enhanced. The graphs are hard to read, and the legends are not clear. I would suggest the authors to use inserts as employed in Fig 5 to clarify the experimental groups. Fig 1: It is not clear on how many animals are in each group. Especially the A section with three bars is confusing. Are there three groups of animals? What are the differences between these three bars if they are all groups with ligation before treatment? If these bars are representing relative brain swelling, this graph reads as brain volume is increasing in volume as almost twice of the original volume. Rephrasing or regraphing is required. Fig 2: Instead of A, B, C, the authors can type the timeline. With an inset they can clarify the treatment. These changes will help the reader grasp the results with considerably less difficulty. “Brain swelling” terminology should be addressed as Fig 1. Fig 5: According to this graph there are negative blood pressure values. Do the authors mean “Change in blood pressure” in the title? Rephrasing or regraphing is required. Inset representing the color of treatment is useful. Fig 6: This graph is way bigger than the others. I would recommend making the graphs similar size. Also, presenting the mass values in mg will be a better choice. Figs 8, 9 and 10: Inset representing the color of treatment would be useful.
  3. Figures should be enhanced. Although the authors have a series of images to present, the presentation is lacking finesse and it is hard to concentrate on the important details. Figs3 and 4: Reader should be oriented to the image with notations. For example, anterior, posterior. The panels can have details typed as “before treatment”, “after treatment”, “saline”, “BPC 157”. Arrows and other markings should be used to relay the messages. Fig 3: It is very hard to get oriented to this figure and get the information presented. It is also very hard to understand the pictures and see the brain swelling that is described with an extremely traumatized brain background that looks globally hemorrhagic. I would recommend presenting only panels C and D in this figure. Fig 4. The panels should be annotated with letters only (either capital or lowercase, without numbers) and a small title should be included in the panels for the summary. For example, current E1 can be titled as “Duodenum - No Treatment” or “Duodenum - Vehicle” (it is not clear in the legend) and current F1 can be titled as “Duodenum - BPC 157”. Arrows and other markers should be used to capture attention to lesions. Figs 12-15: Histological sections should always be presented with scale bars. They should have inserts and markings for clarification.
  4. Authors should not address the animals that had gone sinus ligation as “control”. These animals can be referred as “vehicle treated” animals if they received saline injections or applications. There is a need for “sham” group. A possible approach could be performing the same surgery and passing the suture but not ligating the sinus as tight as to produce thrombosis. Since this surgery looks invasive, it needs to be made clear that the effects are from the occlusion itself rather than craniotomy and hemorrhage. Also, under the Methods section, the surgical protocol is described as “making two burr holes and ligating the sinus to the bone”. However, In Fig 3, it looks like the dorsal skull is removed. Authors need to clarify the used method.
  5. Alternative thrombosis methods can be used. For example, FeCl3 induced thrombosis (Karatas H, Erdener SE, Gursoy-Ozdemir Y, et al. Thrombotic distal middle cerebral artery occlusion produced by topical FeCl(3) application: a novel model suitable for intravital microscopy and thrombolysis studies. J Cereb Blood Flow Metab. 2011;31(6):1452-1460.) is less invasive than suture ligation. Authors are recommended to discuss the choice of their method.
  6. Did the authors consider using microglia specific staining? If not, what is the reasoning?

Minor:

  1. There are many editorial corrections required. Size of the letters are bigger in some sentences. For example, Pg 2, “sinus thrombosis [22,23] and”
  2. Pg 5. “warmed Omnipaque”. What is the temperature?
  3. Pg 7. Is “Stomach, duodenum, liver, spleen, ascites presentation” a title?
  4. Pg 16. “Illustrative is also the rapid worsening.” What does this sentence mean?

Author Response

Dear Editor,

Thank you very much for your kind letter and suggestions and comments given by your reviewers.

The following comments were given by the reviewers.

Reviewer 1

Open Review

English language and style

( ) Extensive editing of English language and style required
( ) Moderate English changes required
(x) English language and style are fine/minor spell check required
( ) I don't feel qualified to judge about the English language and style

Yes

Can be improved

Must be improved

Not applicable

Does the introduction provide sufficient background and include all relevant references?

(x)

( )

( )

( )

Is the research design appropriate?

( )

(x)

( )

( )

Are the methods adequately described?

( )

(x)

( )

( )

Are the results clearly presented?

( )

( )

(x)

( )

Are the conclusions supported by the results?

( )

(x)

( )

( )

Comments and Suggestions for Authors

In this study, Gojkovic et al. presented the vascular recruitment and the effects of pentadecapeptide BPC 157 therapy and the permanent occlusion of the superior sagittal sinus in rat. Although the study is very elaborate and would be useful for researchers in vascular physiology area, the presentation of the data could be dramatically improved. There is a need for sham group. Also, the authors can increase the impact of their study by introducing another model of sagittal sinus thrombosis that is less invasive. Proposed changes are listed in the following section.

Major:

  1. Although the detail oriented study is well appreciated, the proposed publication is too long for smooth reading. I would suggest that the authors condense the introduction and discussion to emphasize the priority of the study. Also, the results should be rephrased to point to the relevant figure. Methods section can be compressed, and details can be submitted as the supplementary material. These changes will increase the reading flow.
  2. Data presentation should be enhanced. The graphs are hard to read, and the legends are not clear. I would suggest the authors to use inserts as employed in Fig 5 to clarify the experimental groups. Fig 1: It is not clear on how many animals are in each group. Especially the A section with three bars is confusing. Are there three groups of animals? What are the differences between these three bars if they are all groups with ligation before treatment? If these bars are representing relative brain swelling, this graph reads as brain volume is increasing in volume as almost twice of the original volume. Rephrasing or regraphing is required. Fig 2: Instead of A, B, C, the authors can type the timeline. With an inset they can clarify the treatment. These changes will help the reader grasp the results with considerably less difficulty. “Brain swelling” terminology should be addressed as Fig 1. Fig 5: According to this graph there are negative blood pressure values. Do the authors mean “Change in blood pressure” in the title? Rephrasing or regraphing is required. Inset representing the color of treatment is useful. Fig 6: This graph is way bigger than the others. I would recommend making the graphs similar size. Also, presenting the mass values in mg will be a better choice. Figs 8, 9 and 10: Inset representing the color of treatment would be useful.
  3. Figures should be enhanced. Although the authors have a series of images to present, the presentation is lacking finesse and it is hard to concentrate on the important details. Figs3 and 4: Reader should be oriented to the image with notations. For example, anterior, posterior. The panels can have details typed as “before treatment”, “after treatment”, “saline”, “BPC 157”. Arrows and other markings should be used to relay the messages. Fig 3: It is very hard to get oriented to this figure and get the information presented. It is also very hard to understand the pictures and see the brain swelling that is described with an extremely traumatized brain background that looks globally hemorrhagic. I would recommend presenting only panels C and D in this figure. Fig 4. The panels should be annotated with letters only (either capital or lowercase, without numbers) and a small title should be included in the panels for the summary. For example, current E1 can be titled as “Duodenum - No Treatment” or “Duodenum - Vehicle” (it is not clear in the legend) and current F1 can be titled as “Duodenum - BPC 157”. Arrows and other markers should be used to capture attention to lesions. Figs 12-15: Histological sections should always be presented with scale bars. They should have inserts and markings for clarification.
  4. Authors should not address the animals that had gone sinus ligation as “control”. These animals can be referred as “vehicle treated” animals if they received saline injections or applications. There is a need for “sham” group. A possible approach could be performing the same surgery and passing the suture but not ligating the sinus as tight as to produce thrombosis. Since this surgery looks invasive, it needs to be made clear that the effects are from the occlusion itself rather than craniotomy and hemorrhage. Also, under the Methods section, the surgical protocol is described as “making two burr holes and ligating the sinus to the bone”. However, In Fig 3, it looks like the dorsal skull is removed. Authors need to clarify the used method.
  5. Alternative thrombosis methods can be used. For example, FeCl3 induced thrombosis (Karatas H, Erdener SE, Gursoy-Ozdemir Y, et al. Thrombotic distal middle cerebral artery occlusion produced by topical FeCl(3) application: a novel model suitable for intravital microscopy and thrombolysis studies. J Cereb Blood Flow Metab. 2011;31(6):1452-1460.) is less invasive than suture ligation. Authors are recommended to discuss the choice of their method.
  6. Did the authors consider using microglia specific staining? If not, what is the reasoning?

Minor:

  1. There are many editorial corrections required. Size of the letters are bigger in some sentences. For example, Pg 2, “sinus thrombosis [22,23] and”
  2. Pg 5. “warmed Omnipaque”. What is the temperature?
  3. Pg 7. Is “Stomach, duodenum, liver, spleen, ascites presentation” a title?
  4. Pg 16. “Illustrative is also the rapid worsening.” What does this sentence mean?

Reviewer 2

Open Review

English language and style

( ) Extensive editing of English language and style required
(x) Moderate English changes required
( ) English language and style are fine/minor spell check required
( ) I don't feel qualified to judge about the English language and style

Yes

Can be improved

Must be improved

Not applicable

Does the introduction provide sufficient background and include all relevant references?

( )

(x)

( )

( )

Is the research design appropriate?

(x)

( )

( )

( )

Are the methods adequately described?

( )

(x)

( )

( )

Are the results clearly presented?

( )

(x)

( )

( )

Are the conclusions supported by the results?

(x)

( )

( )

( )

Comments and Suggestions for Authors

The author tested the gastric peptide BCP 157. The work was extensive and challenging to read due to different techniques and similar words in the paragraphs. Here are my suggestions to improve the paper quality.

Major- My main concern is an almost identical result between the different routes of administration. There is no mention of the bioavailability of this peptide or comment about the used dose. It looks like that the maximum effect is reached at 10ng/Kg at any route. What intrigues me is that 10ng/kg intragastric (I imagine that they used gavage) gets the brain in one minute. I have no problems with liposoluble drugs injected IV, but I have my concerns about peptides.  From my perspective, they are showing that the minimal effective drug concentration reaches the brain ate 1 minute, because there is no change in time and dose-cocentration. Still, again there is no discussion about that.

 It is complicated to analyze the results when they are in percentage and not in absolute numbers. What is the reason to do that?

Minor-

Abstract

Please rephrase Assessments until duodenal. It is confusing

Pressure- what type of pressure, blood-pressure intracranial pressure, mean blood pressure, systolic, diastolic?

Introduction

Last paragraph: It is an explanation for methods. You repeat the same information.

Please change sacrifice for euthanized. (several times)

2.7 Change Microscopy to Histology (H&E).  It is more accurate.

Please give more details to H&E even though it is a well-known technique.  

No size bar in the images. Please add a ruler with size in um.

Please add arrows to show what you would like to focus the reader's attention.

None of the figures has the power. I would like to see it.

Figure 13. Less than 5 cells microglia cells per mm2, cm2.  It will be of tremendous help to show if this microglia is at M2 -anti-inflammatory state. There are different markers to verify M1 and M2 states. The migration could be less critical if the cells are still inflammatory. This should be done at 48 hours.

In the figure of Blood pressure. What are you measuring? Mean blood pressure? Diastolic, systolic?

To the  comments given by the reviewers, see our arguments and reply.

Reviewer 1

General

We highly appreciate the comments given by the reviewer. The first part of his/her comments is related to the presentation of the Introduction and Discussion, and presentation of the Figures. In this context, we fully appreciated his/her comments about the Introduction and Discussion, which are both completely rewritten and focused to the prime point of this study, central venous occlusion, occlusion of the superior sagittal sinus, and consequent noxious syndrome, and the therapy background for the stable pentadecapeptide BPC 157, seen previously in the resolving of the peripheral venous syndromes (i.e., inferior caval vein syndrome (infrarenal occlusion of the inferior caval vein), Pringle maneuver ischemia, reperfusion (portal triad temporary occlusion) and Budd Chiari syndrome (suprahepatic occlusion of the inferior caval vein) with application of the stable gastric pentadecapeptide BPC 157), and evidence about the activation of the bypassing loops, centrally and peripherally.  Likewise, we fully appreciate the comments given for the Figures, and all of the pointed Figures are markedly improved, and accommodated with the requests of the Reviewer 1. In particular, histology figures are now given with the requested scale. Of note, the legends are also accordingly rewritten.

We also appreciated the concern of the Reviewer about the calvariectomy and severity of the used procedure, and thereby need for further sham procedure(s). However, we should emphasize that calvariectomy as procedure is used in the therapy of the increased intracranial pressure (as laparotomy for the abdominal compartment syndrome therapy), and thereby unlikely to aggravate (and interfere) with the circumstances produced by the occlusion of the superior sagittal sinus. Besides, we mentioned in the methods text that calvariectomy procedure (including complete calvariectomy) did not produce by itself any harm. Besides, the  procedure proposed by the Reviewer would use calvariectomy as well. In addition, it seems to us that procedure proposed as sham procedure – does not match with the procedure used in the present study. Narrowing of the blood vessel (a procedure frequently used for the portal hypertension induction) means a sustained worsening – while occlusion of the superior sagittal sinus include an abrupt procedure, and immediate presentation of the whole syndrome. The Karakas et and coworkers procedure with FeCl3 includes in addition to the calvarial window, a direct contact, i.e., 3 min long, with the vessel (we should also indicate that this study does not cover superior sagittal sinus). Thus, it seems to us that the procedure used in the present study is advantageous (instant occlusion of the superior sagittal sinus (vessel) means that the prime injury is highly reproducible, site of the injury easy to be identified, and not interrupted course of the noxious events ascertained). Finally, providing that Reviewer had concern about the effect of BPC 157 on thrombosis formation, it should be noted that this point was addressed in three recent peripheral venous occlusion studies, as well as before, with abdominal aorta anastomosis. In the venous occlusion studies, the thrombosis formation was clearly ascribed to the vessel occlusion (as it is the case in the present study).  Thus, it seems to us that the use of the occlusion of the superior sagittal sinus as suited method is fully justified. Also, considering the Reviewer’s proposal about the use of another model of sinus sagittal thrombosis, but less invasive, it is quite safe to claim that the stronger model would cover the milder model as well, but not vice versa. This point is now specially addressed in the revised Discussion (see paragraph 3)

Most certainly, the sinus ligated to the bone ascertained definitive occlusion,   skull bone overlying the sinus eliminated the risk of sinus tear. Thereby, such ligation method of the superior sagittal sinus means permanent occlusion, permanent alteration of blood flow, and permanent increased pressure in the superior sagittal sinus. In this, calvariectomy and/or laparotomy (used to assess intracranial (superior sagittal sinus), portal, inferior caval vein and aortal pressure, and brain swelling and organs lesions and vessels presentation) since used in the therapy to counteract increased intracranial pressure and abdominal compartment syndrome [56,57], did not further contribute to the worst circumstances created by the occlusion of the superior sagittal sinus. Thereby,  the evidenced high superior sagittal sinus, portal and caval hypertension and aortal hypotension are along with the rapid worsening that would appear along with the superior sagittal sinus occlusion. As before in the peripheral venous occlusion studies [20-27], clear prime injury site (i.e., occlusion by ligation), not removable, would continuously perpetuate injurious course, a point that can be hardly secured otherwise (i.e., FeCl3 thrombosis) [58]. 

Besides, we should indicate that the main point in the study is the organization of the bypassing loop to compensate occluded superior sagittal sinus. In our view, this may be possible only with the clear prime injury (ligation-occlusion) – which could be not removed – therefore, ascertaining an uninterrupted noxious course as an essential point for the further research and possible therapy application.

Specific comments

In this study, Gojkovic et al. presented the vascular recruitment and the effects of pentadecapeptide BPC 157 therapy and the permanent occlusion of the superior sagittal sinus in rat. Although the study is very elaborate and would be useful for researchers in vascular physiology area, the presentation of the data could be dramatically improved. There is a need for sham group. Also, the authors can increase the impact of their study by introducing another model of sagittal sinus thrombosis that is less invasive. Proposed changes are listed in the following section.

Acknowledged. However, it should be mentioned that the occlusion of the superior sagittal sinus is used as a model of the permanent central venous occlusion and as such, its possible significance for thrombosis development may override any other less invasive model. In this context, we should emphasize that the occlusion of the superior sagittal sinus follows the value of the similar occlusion of the inferior caval vein, infrarenal, and suprahepatic, superior anterior pancreaticoduodenal vein, left colic artery and vein – and thereby, the effect of the permanent venous occlusion itself appears to be established beyond any doubt.

Major:

  1. Although the detail oriented study is well appreciated, the proposed publication is too long for smooth reading. I would suggest that the authors condense the introduction and discussion to emphasize the priority of the study. Also, the results should be rephrased to point to the relevant figure. Methods section can be compressed, and details can be submitted as the supplementary material. These changes will increase the reading flow. Introduction and Discussion are completely rewritten, focused on the priority of the study (i.e., permanent central venous occlusion, and the therapy) better emphasized. In the Results, all of the appointed Figures are improved.
  2. Data presentation should be enhanced. The graphs are hard to read, and the legends are not clear. I would suggest the authors to use inserts as employed in Fig 5 to clarify the experimental groups. Fig 1: It is not clear on how many animals are in each group. Especially the A section with three bars is confusing. Are there three groups of animals? What are the differences between these three bars if they are all groups with ligation before treatment? If these bars are representing relative brain swelling, this graph reads as brain volume is increasing in volume as almost twice of the original volume. Rephrasing or regraphing is required. Fig 2: Instead of A, B, C, the authors can type the timeline. With an inset they can clarify the treatment. These changes will help the reader grasp the results with considerably less difficulty. “Brain swelling” terminology should be addressed as Fig 1. Fig 5: According to this graph there are negative blood pressure values. Do the authors mean “Change in blood pressure” in the title? Rephrasing or regraphing is required. Inset representing the color of treatment is useful. Fig 6: This graph is way bigger than the others. I would recommend making the graphs similar size. Also, presenting the mass values in mg will be a better choice. Figs 8, 9 and 10: Inset representing the color of treatment would be useful. Fig. 1 and Fig. 2 are accordingly modified, and legend accommodated. The occlusion of the superior sagittal sinus may induce considerable brain swelling. Fig. 6 appears to be same as the Fig. 1, and its size would be technically easy to accommodate.
  1. Figures should be enhanced. Although the authors have a series of images to present, the presentation is lacking finesse and it is hard to concentrate on the important details. Figs3 and 4: Reader should be oriented to the image with notations. For example, anterior, posterior. The panels can have details typed as “before treatment”, “after treatment”, “saline”, “BPC 157”. Arrows and other markings should be used to relay the messages. Fig 3: It is very hard to get oriented to this figure and get the information presented. It is also very hard to understand the pictures and see the brain swelling that is described with an extremely traumatized brain background that looks globally hemorrhagic. I would recommend presenting only panels C and D in this figure. Fig 4. The panels should be annotated with letters only (either capital or lowercase, without numbers) and a small title should be included in the panels for the summary. For example, current E1 can be titled as “Duodenum - No Treatment” or “Duodenum - Vehicle” (it is not clear in the legend) and current F1 can be titled as “Duodenum - BPC 157”. Arrows and other markers should be used to capture attention to lesions. Figs 12-15: Histological sections should always be presented with scale bars. They should have inserts and markings for clarification. Acknowledged. Fig. 3 and Fig. 4 are completely changed, and hopefully adequately improved. The legends are rewritten. Histological sections are presented with scale bars.  

  1. Authors should not address the animals that had gone sinus ligation as “control”. These animals can be referred as “vehicle treated” animals if they received saline injections or applications. There is a need for “sham” group. A possible approach could be performing the same surgery and passing the suture but not ligating the sinus as tight as to produce thrombosis. Since this surgery looks invasive, it needs to be made clear that the effects are from the occlusion itself rather than craniotomy and hemorrhage. Also, under the Methods section, the surgical protocol is described as “making two burr holes and ligating the sinus to the bone”. However, In Fig 3, it looks like the dorsal skull is removed. Authors need to clarify the used method. Acknowledged. Considering the point of the “invasive” surgery, we should claim that the surgery is not more invasive than it was the case with the Pringle maneuver or Budd-Chiari syndrome – and thereby, the used surgery could not interfere with the conclusion that the thrombosis as an outcome could be certainly ascribed to the ligation (occlusion) itself. Considering the Fig. 3, the Fig. 3 could be not used as an argument in favor of the method incapability to assess further thrombosis development, since see Method, the calvarial window was used just for the brain swelling recording.

  1. Alternative thrombosis methods can be used. For example, FeCl3 induced thrombosis (Karatas H, Erdener SE, Gursoy-Ozdemir Y, et al. Thrombotic distal middle cerebral artery occlusion produced by topical FeCl(3) application: a novel model suitable for intravital microscopy and thrombolysis studies. J Cereb Blood Flow Metab. 2011;31(6):1452-1460.) is less invasive than suture ligation. Authors are recommended to discuss the choice of their method. Acknowledged. It seems to safe to speculate that the stronger method occlusion of the superior sagittal sinus may override any other method that would not use occlusion. We should emphasize that counteraction of the thrombosis development was reported in few our previous studies. In the reply to the introductory remark of the reviewer, the argumentation is already given. This point is now clearly emphasized (Discussion, paragraph 3). Considering the point of the “invasive” surgery, we should claim that the surgery is not more invasive than it was the case with the Pringle maneuver or Budd-Chiari syndrome – and thereby the thrombosis as an outcome could be certainly ascribed to the ligation (occlusion) itself. Considering the Fig. 3, the Fig. 3 could be not used as an argument in favor of the method incapability to assess further thrombosis development, since see Method, the craniotomy fenestration was used just for the brain swelling recording.
  2. Did the authors consider using microglia specific staining? If not, what is the reasoning? Acknowledged. See 2.7.1. section in Materials and methods, Figure 13 and Figure 14 in Results,and in Discussion, paragraph 5.

Minor:

  1. There are many editorial corrections required. Size of the letters are bigger in some sentences. For example, Pg 2, “sinus thrombosis [22,23] and” Acknowledged and corrected.
  2. Pg 5. “warmed Omnipaque”. What is the temperature? 370C
  3. Is “Stomach, duodenum, liver, spleen, ascites presentation” a title? Acknowledged and corrected.
  4. Pg 16. “Illustrative is also the rapid worsening.” What does this sentence mean? Text is fully rewritten.

Reviewer 2

The author tested the gastric peptide BCP 157. The work was extensive and challenging to read due to different techniques and similar words in the paragraphs. Here are my suggestions to improve the paper quality.

Major- My main concern is an almost identical result between the different routes of administration. There is no mention of the bioavailability of this peptide or comment about the used dose. It looks like that the maximum effect is reached at 10ng/Kg at any route. What intrigues me is that 10ng/kg intragastric (I imagine that they used gavage) gets the brain in one minute. I have no problems with liposoluble drugs injected IV, but I have my concerns about peptides.  From my perspective, they are showing that the minimal effective drug concentration reaches the brain ate 1 minute, because there is no change in time and dose-cocentration. Still, again there is no discussion about that.

 Acknowledged. We fully appreciate the concern of the reviewer considering the intragastric application. Of note, BPC 157 is effective given intragastrically, as well as given perororally in drinking water. To see this issue, we would suggest to go through our reviews (indicated as ref. 5-19). Now, this point is fully appreciated, first in Introduction.

Also, in the stroke-studies, given after  reperfusion initiation, after carotid arteries clamping,   BPC 157 abrogated hippocampal ischemia/reperfusion injuries in rats [30]. Besides, BPC 157 may participate in the brain-gut and gut-brain axis function [9], exert particular effects, when given peripherally [9] (i.e., release of the serotonin in the specific brain areas (i.e., nigrostriatum) [31], opposes  the schizophrenia-like positive symptoms models [32], counteracts various encephalopathies [30,33-41]). 

Accordingly, with BPC 157 therapy in the rats with the permanently occluded superior sagittal sinus, rapid upgrading of the bypassing pathways, both centrally and peripherally,  would resolve vessel occlusion disturbances, both centrally and peripherally [20–27]. The study includes both the very early and prolonged periods. To cover the full injury course, both the acute and prolonged periods, the supporting evidence goes with the various time points (i.e., at 1 min, 15 min, 24 h or 48 h ligation-time) for the application of the BPC 157 therapy. Local application at the swollen brain means a direct effect; intraperitoneal or intragastric administrations mean a systemic effect, µg- and ng-regimens mean a common beneficial effect. Conceptually, intragastric application benefits BPC 157 importance as an original cytoprotective anti-ulcer peptide (i.e., epithelium, endothelium maintenance and protection) [5-19]. It is  resistant and stable in the human gastric juice (more than 24h [42]), acting as a membrane stabilizer, counteracting leaky gut syndrome, as a particular target [19],  distinctive from the standard peptide growth factors [16] (i.e., standard peptide growth factors are rapidly destroyed in human gastric juice [42]),  with particular molecular pathways involved [15,19,20,28,43-50]. Likely, BPC 157 is controlling VEGF- and NO-pathways [43,47]. BPC 157 also immediately triggered the internalization of VEGFR2 and subsequently activated the phosphorylation of VEGFR2, Akt, and eNOS signal pathway without the need of other known ligands or shear stress [47]. Finally, in addition to this particular effect [47],   there is a direct effect on vasomotor tone (i.e., BPC 157 counteracts both L-NAME-induced hypertension and L-arginine-induced hypotension, and there is a specific activation of Src-Caveolin-1-endothelial nitric oxide synthase (eNOS) pathway [43].  There, the activated “bypassing key” appeared as an outbreak of the original cytoprotection agent’s activity of  “the endothelium protection → epithelium protection” [1,7,8]. The additional equation  “endothelium maintenance → epithelium maintenance = blood vessel recruitment and activation (“running”) towards the site of injury, also described as “bypassing” occlusion via alternative ways” [7,8], was shown in the BPC 157 therapy effect in the previous peripheral vascular occlusion studies [20-27].

Discussion, paragraph 2 and paragraph 4.

Vice versa, confronted with the permanent central venous occlusion, the therapy effect is starting centrally, from the occluded superior sagittal sinus, as a prime noxious event (note, instead increased positive pressure, BPC 157-rats presented the  negative pressure values close to those noted  in the superior sagittal sinus of the healthy rats). Evidently, it successfully extended the previous BPC 157’s “bypassing key”, activated specific collateral pathways that successfully overwhelmed peripheral major veins occlusion and largely attenuated/eliminated the otherwise deadly peripheral venous occlusion syndromes, abrogated portal and caval hypertension and aortal hypotension [26,27].  As continuation, we revealed the novel, applicable therapy evidence, centrally, “bypassing key”, activated specific collateral pathways, and BPC 157 effectiveness, µg-ng, regimens, given topically at the swollen brain, intraperitoneally or intragastrically, as an early or delayed application. The shunts are through ophthalmic vein, angularis vein, facial anterior and posterior vein, and facial vein, or through cerebri superior veins, sinus cavernosus, sinus petrosus superior and inferior, sinus transversus, through jugular external vein, subclavia vein through superior caval vein. Thereby, there was rapid attenuation of the brain swelling. The rapid elimination of the increased pressure in the ligated superior sagittal sinus, rapid elimination of the severe portal and caval hypertension and aortal hypotension, and rapid collateral vessels recruitment, abrogated venous and arterial thrombosis and recovery of the organs lesions, consistently occurred. Together, the beneficial action is going on both  centrally and peripherally, overwhelming both central and peripheral harms of the  permanent central venous occlusion.  There is the activated azygos vein pathway, left superior caval vein-azygos vein-inferior caval vein, and, consequently, congested inferior caval vein and superior mesenteric vein became decongested reflecting elimination of the otherwise severe caval and portal hypertension.  This occurred as it had been noted in Budd-Chiari syndrome BPC 157 therapy,  with either intraperitoneal or per-oral application [27], indicative for the syndrome of the occluded superior sagittal sinus.

 It is complicated to analyze the results when they are in percentage and not in absolute numbers. What is the reason to do that? Acknowledged. This implies the assessment performed in vivo, and thereby, using of the % of the normal healthy rats.

Thereby, as mentioned, it may be relevant the described immediate gross disappearance of the swollen brain presentation, and microscopically only a few karyopyknotic neurons upon administration of the BPC 157 regimens, given locally at the swollen brain, or intragastrically or intraperitoneally, thus, evidencing a consistent BPC 157 effect, as shown by the direct recording. The attenuated were progressive cerebral edema, which may the net capillary filtration increase, and intracerebral and subarachnoid hemorrhage, which may additionally compromise the brain tissue [60].  Interestingly, it may be that these findings are showing that the minimal effective drug concentration very rapidly reaches the brain (i.e., at 1 minute), because there is no change in time and dose-concentration. On the other hand, in particular considering BPC 157 intragastric application,  these findings are along with the above mentioned BPC 157 supposed conceptual significance as an original cytoprotective anti-ulcer peptide resistant to human gastric juice [5-19], and essential part of function of brain-gut axis and gut-brain axis.

Minor-

Abstract

Please rephrase Assessments until duodenal. It is confusing Acknowledged and corrected.

Pressure- what type of pressure, blood-pressure intracranial pressure, mean blood pressure, systolic, diastolic? Acknowledged mean blood pressure.

Introduction Acknowledged, Introduction is completely rewritten (see our reply to the comments of the Rev. 1).

Last paragraph: It is an explanation for methods. You repeat the same information. Acknowledged. Removed, and included in Discussion considering the comments given by the Reviewer 1.

Please change sacrifice for euthanized. (several times) Acknowledged and corrected.

2.7 Change Microscopy to Histology (H&E).  It is more accurate. Acknowledged.

Please give more details to H&E even though it is a well-known technique.  Acknowledged and corrected.

No size bar in the images. Please add a ruler with size in um. Acknowledged and corrected.

Please add arrows to show what you would like to focus the reader's attention. Acknowledged and corrected.

None of the figures has the power. I would like to see it.

Figure 13. Less than 5 cells microglia cells per mm2, cm2.  It will be of tremendous help to show if this microglia is at M2 -anti-inflammatory state. There are different markers to verify M1 and M2 states. The migration could be less critical if the cells are still inflammatory. This should be done at 48 hours. Acknowledged and corrected (see our reply to the comment of the Rev. 1)

In the figure of Blood pressure. What are you measuring? Mean blood pressure? Diastolic, systolic?

Acknowledged. Mean blood pressure.

In summary, we hope that you will find this manuscript to be adequately improved and finally suited for final presentation in your distinguished journal.

Looking forward to hearing from you

Sincerely

Predrag Sikiric, MD, PhD

Professor

Reviewer 2 Report

The author tested the gastric peptide BCP 157. The work was extensive and challenging to read due to different techniques and similar words in the paragraphs. Here are my suggestions to improve the paper quality.

Major- My main concern is an almost identical result between the different routes of administration. There is no mention of the bioavailability of this peptide or comment about the used dose. It looks like that the maximum effect is reached at 10ng/Kg at any route. What intrigues me is that 10ng/kg intragastric (I imagine that they used gavage) gets the brain in one minute. I have no problems with liposoluble drugs injected IV, but I have my concerns about peptides.  From my perspective, they are showing that the minimal effective drug concentration reaches the brain ate 1 minute, because there is no change in time and dose-cocentration. Still, again there is no discussion about that.

 It is complicated to analyze the results when they are in percentage and not in absolute numbers. What is the reason to do that?

Minor-

Abstract

Please rephrase Assessments until duodenal. It is confusing

Pressure- what type of pressure, blood-pressure intracranial pressure, mean blood pressure, systolic, diastolic?

Introduction

Last paragraph: It is an explanation for methods. You repeat the same information.

Please change sacrifice for euthanized. (several times)

2.7 Change Microscopy to Histology (H&E).  It is more accurate.

Please give more details to H&E even though it is a well-known technique.  

No size bar in the images. Please add a ruler with size in um.

Please add arrows to show what you would like to focus the reader's attention.

None of the figures has the power. I would like to see it.

Figure 13. Less than 5 cells microglia cells per mm2, cm2.  It will be of tremendous help to show if this microglia is at M2 -anti-inflammatory state. There are different markers to verify M1 and M2 states. The migration could be less critical if the cells are still inflammatory. This should be done at 48 hours.

In the figure of Blood pressure. What are you measuring? Mean blood pressure? Diastolic, systolic?

Author Response

(The authors gave the same response as above.)

Round 2

Reviewer 2 Report

The authors answered all my concerns. No further modifications are necessary.